# MyD88-dependent influx of monocytes and neutrophils impairs lymph node B cell responses to chikungunya virus infection via Irf5, Nos2 and Nox2

Mary K. McCarthy[1], Glennys V. Reynoso[2], Emma S. Winkler[3], Matthias Mack[4], Michael S. Diamond[3,5,6,7], Heather D. Hickman[2], Thomas E. Morrison[1]*

1 Department of Immunology and Microbiology, University of Colorado School of Medicine, Aurora, Colorado, United States of America, 2 Viral Immunity and Pathogenesis Unit, Laboratory of Clinical Microbiology and Immunology, National Institutes of Allergy and Infectious Diseases, NIH, Bethesda, Maryland, United States of America, 3 Department of Medicine, Washington University School of Medicine, St. Louis, Missouri, United States of America, 4 Regensburg University Medical Center, Regensburg, Germany, 5 Department of Pathology and Immunology, Washington University School of Medicine, St. Louis, Missouri, United States of America, 6 Department of Molecular Microbiology, Washington University School of Medicine, St. Louis, Missouri, United States of America, 7 The Andrew M. and Jane M. Bursky Center for Human Immunology and Immunotherapy Programs, Washington University School of Medicine, St. Louis, Missouri, United States of America

* thomas.morrison@cuanschutz.edu

**Data Availability Statement:** All relevant data are within the manuscript and its Supporting Information files.

## Abstract

Humoral immune responses initiate in the lymph node draining the site of viral infection (dLN). Some viruses subvert LN B cell activation; however, our knowledge of viral hindrance of B cell responses of important human pathogens is lacking. Here, we define mechanisms whereby chikungunya virus (CHIKV), a mosquito-transmitted RNA virus that causes outbreaks of acute and chronic arthritis in humans, hinders dLN antiviral B cell responses. Infection of WT mice with pathogenic, but not acutely cleared CHIKV, induced MyD88-dependent recruitment of monocytes and neutrophils to the dLN. Blocking this influx improved lymphocyte accumulation, dLN organization, and CHIKV-specific B cell responses. Both inducible nitric oxide synthase (iNOS) and the phagocyte NADPH oxidase (Nox2) contributed to impaired dLN organization and function. Infiltrating monocytes expressed iNOS through a local IRF5- and IFNAR1-dependent pathway that was partially TLR7-dependent. Together, our data suggest that pathogenic CHIKV triggers the influx and activation of monocytes and neutrophils in the dLN that impairs virus-specific B cell responses.

## Author summary

Elucidating mechanisms by which viruses subvert B cell immunity and establish persistent infection is essential for the development of new therapeutic strategies against chronic viral infections. The humoral immune response initiates in the lymph node draining the site of viral infection. However, how persistent viruses evade B cell responses is poorly

**Funding:** This work was supported by Public Health Service grants U19 AI109680 (T.E.M.), R01 AI141436 (T.E.M. and M.S.D.), R01 AI089591 and R01 AI114816 (M.S.D), and F32 AI122463 (M.K. M.) from the National Institute of Allergy and Infectious Diseases (https://www.niaid.nih.gov/). H.D.H. is supported by the Intramural Research Program of NIAID, NIH (https://www.niaid.nih.gov/). The funders had no role in study design, data collection and analysis, decision to publish, or preparation of the manuscript.

**Competing interests:** I have read the journal's policy and the authors of this manuscript have the following competing interests: M.S.D. is a consultant for Inbios and Atreca and on the Scientific Advisory Board of Moderna. The other authors declare no competing financial interests.

understood. In this study, we find that infection with pathogenic, persistent chikungunya virus triggers rapid recruitment of neutrophils and monocytes to the draining lymph node, which impair structural organization, lymphocyte accumulation, and downstream virus-specific B cell responses that are important for control of infection. This work enhances our understanding of the pathogenesis of acute and chronic CHIKV disease and highlights how local innate immune responses in draining lymphoid tissue dictate the effectiveness of downstream adaptive immunity.

## Introduction

Draining lymphoid organs have a critical role in the initiation of effective adaptive immune responses to vaccines and pathogens. B cells and virus-specific antibodies (Abs) are important arms of the adaptive immune response for limiting viral dissemination and controlling viral infections. The antiviral Ab response is initiated in the draining lymph node (dLN) through a series of molecular interactions and cellular movements that require intact lymphoid tissue architecture and function. Naïve B cells enter the dLN through high endothelial venules (HEVs) and then migrate to defined follicles subjacent to the dLN subcapsular sinus (SCS) [1]. B cells acquire viral antigen through interactions with SCS macrophages which readily acquire viral antigens borne by the afferent lymphatics [2]. After activation, B cells migrate to the borders of the B cell follicle and to the interfollicular region of the dLN to encounter CD4+ T cells. Some of these activated, proliferating B cells go on to form plasmablasts, while others persist in tight clusters in the follicle, forming germinal centers (GCs) [3, 4].

Some viruses have evolved strategies to subvert humoral immune responses in the dLN. Lymphocytic choriomeningitis virus (LCMV), a prototypical mouse pathogen, impairs Ab responses in the dLN via a number of mechanisms, including the recruitment of inflammatory monocytes that promote B cell apoptosis [5]. Vaccinia virus disrupts the dLN SCS macrophage layer, preventing antigen acquisition and subsequent B cell responses [6]. In humans, HIV infection of CD4+ T cells results in loss of follicular T helper cell responses and impaired Ab production [7].

Arthritogenic alphaviruses, including chikungunya virus (CHIKV), Mayaro virus, and Ross River virus, are emerging public health concerns. CHIKV, in particular, re-emerged in 2004 to cause massive outbreaks of disease affecting millions [8]. Acute CHIKV disease presents with rapid onset of fever and rash, with severe joint swelling and arthralgia. A high proportion of individuals develop chronic arthralgia or arthritis that lasts for months to years after initial infection [9]. Following CHIKV infection, the humoral response reduces acute disease severity and controls viremia [10], and passive transfer of immune serum or monoclonal antibodies protects from fatal CHIKV disease [11–13]. However, B cells and Ab are unable to mediate complete clearance of CHIKV from all sites of infection [14–22]. Thus, chronic CHIKV disease in musculoskeletal tissue may be driven by failure of the adaptive immune response to prevent or control viral persistence.

Clearance of the attenuated CHIKV strain 181/25 from musculoskeletal tissue depends on virus-specific Ab production. In comparison, pathogenic CHIKV strains persist in joint-associated tissue of immunocompetent mice [14, 15], suggesting that the development of virus-specific B cell responses may be impaired during CHIKV infection. Indeed, we previously found that pathogenic CHIKV strains disable immune responses in the dLN of infected mice by impairing naïve lymphocyte entry and expansion of HEVs [23]. Additionally, the dLN exhibited follicular disorganization, with a loss of the B-T cell border and poor GC formation

following pathogenic CHIKV infection. However, the mechanisms leading to architectural disruption and impaired lymphocyte accumulation remained to be defined.

In this study, we find that a MyD88-dependent influx of monocytes and neutrophils to the dLN disrupts structural organization, impairs lymphocyte accumulation, and diminishes downstream antiviral B cell responses to pathogenic CHIKV infection in a manner partially dependent on expression of inducible nitric oxide synthase (iNOS) and the phagocyte NADPH oxidase (Nox2). We also identify a role for local IRF5-driven type I IFN signaling in the dLN that is triggered by TLR7 and one or more additional TLR(s) and promotes iNOS expression in infiltrating monocytes. Overall, our data suggest that the rapid, early influx of monocytes and neutrophils detrimentally impacts the B cell response to pathogenic CHIKV infection through a novel IRF5-dependent mechanism.

## Results

### Pathogenic CHIKV infection triggers recruitment of monocytes and neutrophils to the dLN

In previous studies, we found that infection of mice with the attenuated and acutely cleared CHIKV strain 181/25 [14, 24] resulted in dLN hypertrophy, accumulation of lymphocytes, and formation of GCs [23], all hallmarks of appropriate innate and adaptive immune responses to the infection. In contrast, the dLN of mice infected with pathogenic, persistent CHIKV strains became highly disorganized as early as 3 days post-infection (dpi), and failed to accumulate lymphocytes or develop GCs due in part to disruption of HEV-mediated lymphocyte recruitment [23]. Consistent with the robust response of the dLN to attenuated 181/25 infection, the presence of LNs is essential for clearance of 181/25 infection from joint-associated tissue (**S1 Fig**). To elucidate the mechanisms that lead to the rapid disorganization and function of the dLN following pathogenic CHIKV infection, we characterized cell populations at early times post-infection with pathogenic (parental strain AF15561) or attenuated (derivative strain 181/25) CHIKV, which differ by only five amino acids across the genome [25]. Pathogenic, but not attenuated CHIKV infection, resulted in increased numbers of monocytes (CD11b$^+$Ly6C$^{hi-}$Ly6G$^-$) and neutrophils (CD11b$^{hi}$Ly6C$^+$Ly6G$^+$) in the blood and dLN within 24 h of infection (**S2A–S2C Fig and Fig 1A–1C**). The influx of monocytes and neutrophils into the dLN preceded their arrival in the foot, the site of virus inoculation (**S2D–S2F Fig**), which occurs between 3 and 5 dpi [26]. In contrast, few monocytes and neutrophils were present in the dLNs of mock-infected mice or mice inoculated with the attenuated CHIKV strain (**Fig 1D**). Consistent with our flow cytometric analysis, immunofluorescence staining of dLN sections at 1 dpi detected a large population of Gr-1$^+$ monocytes and neutrophils in the dLNs of mice infected with pathogenic CHIKV. These cells were mostly localized within the SCS and medullary sinuses, with a smaller population of monocytes and neutrophils within the B cell follicles (**Fig 1D**). Thus, during pathogenic, but not attenuated, CHIKV infection, a rapid influx of monocytes and neutrophils into the dLN precedes its extensive disorganization.

### Monocyte and neutrophil influx causes reduced lymphocyte accumulation and dLN disorganization

Pathogenic CHIKV infection results in decreased accumulation of naïve lymphocytes and extensive lymphocyte relocalization in the dLN [23]. Because monocyte and neutrophil infiltration of the dLN preceded the disruption of lymphocyte populations (**Fig 1** and [23]), we hypothesized that accumulation of myeloid cells in the dLN might disrupt its architecture. To prevent the early influx of monocytes and neutrophils into the dLN, we treated mice with a

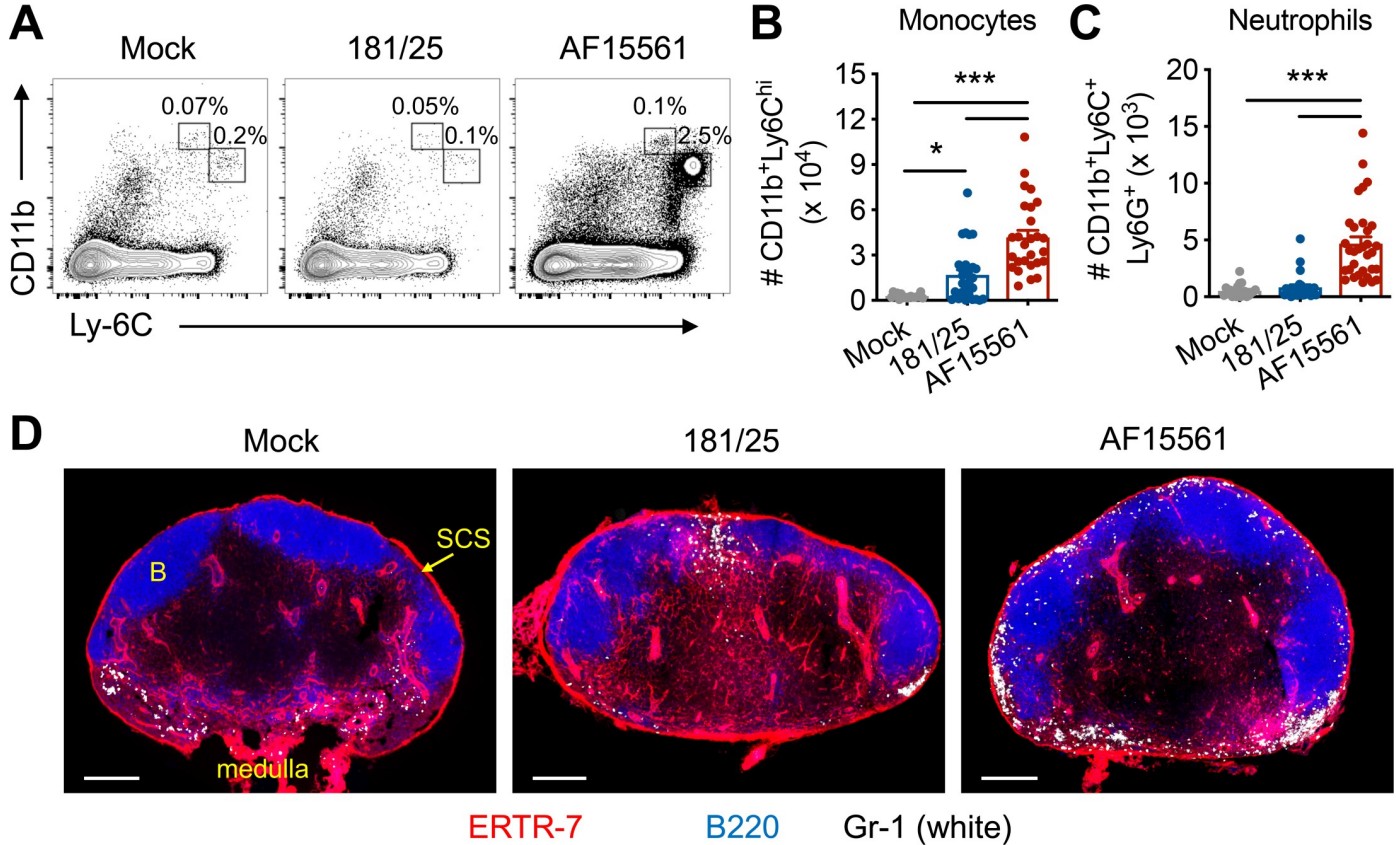

**Fig 1. Myeloid cells infiltrate the dLN early after pathogenic CHIKV infection.** C57BL/6 mice were inoculated with PBS (mock) or $10^3$ plaque forming units (PFU) of CHIKV 181/25 or AF15561 in the left footpad, and the dLN was analyzed at 24 hpi. (**A**) Representative flow cytometry plots and numbers of (**B**) CD11b$^+$Ly6C$^{hi}$ monocytes or (**C**) CD11b$^{hi}$Ly6C$^+$Ly6G$^+$ neutrophils in the dLN. (**D**) Frozen dLN sections were stained for ERTR7$^+$ stromal cells (red), B220$^+$ B cells (blue), or Gr-1$^+$ monocytes and neutrophils (white). Errors bars represent mean ± SEM. Data are combined from or representative of 5 (**A-C**) or 2 (**D**) independent experiments (n = 15–21 (**A-C**) or 2–3 (**D**) mice per group). Statistical significance was determined by one-way ANOVA with Tukey's post-test ($^*$, $P < 0.05$; $^{***}$, $P < 0.001$).

single dose of anti-Gr-1 monoclonal antibody (mAb) the day before inoculation with pathogenic CHIKV, which effectively depleted monocytes and neutrophils from the circulation (**Fig 2A–2C**) and the dLN (**Fig 2D–2F**). Remarkably, a single anti-Gr-1 mAb treatment prior to pathogenic CHIKV infection restored total cell numbers at 5 dpi in the dLN to levels nearly equivalent to those detected during acutely cleared CHIKV infection (**Fig 2G**), an effect that was due principally to changes in B and CD4$^+$ T cell numbers (**Fig 2H and 2I**). CD8$^+$ T cell numbers in anti-Gr-1-treated mice remained lower than in mice infected with the attenuated CHIKV strain (**Fig 2J**). The failure to fully restore CD8$^+$ T cell numbers may reflect some effect of the anti-Gr-1 mAb on CD8$^+$ T cells, some of which express Gr-1 upon activation [27]. Depletion of monocytes and neutrophils prior to attenuated CHIKV infection did not affect total cell numbers in the dLN at 5 dpi (**S3 Fig**).

Pathogenic CHIKV infection causes marked disorganization of lymphocyte populations within the dLN [23], with paracortical relocalization of B cells, diffuse positioning of CD8$^+$ T cells, and loss of a well-defined B-T cell border by 5 dpi. In mice treated with anti-Gr-1 mAb prior to pathogenic CHIKV infection, we observed improved follicular organization and a defined B-T cell border compared with the isotype control group (**Fig 2K**). Together, these data suggest that the early influx of monocytes and neutrophils into the dLN following

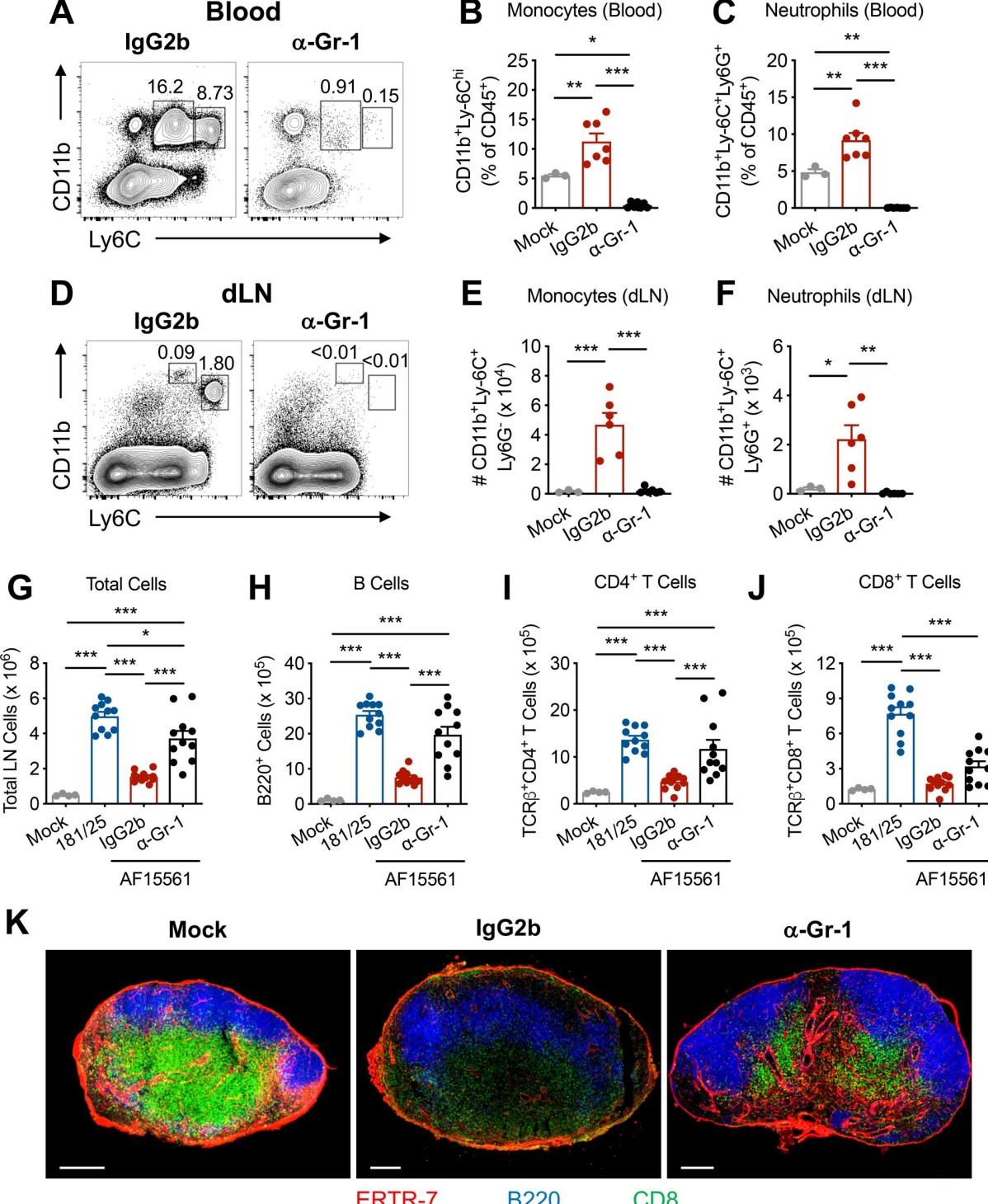

**Fig 2. Delaying the early influx of myeloid cells to the dLN prevents lymphocyte depletion and restores dLN architecture.** C57BL/6 mice were left untreated or treated with 300–500 μg of IgG2b isotype control mAb or anti-Gr-1 mAb via intraperitoneal injection one day prior to inoculation with $10^3$ PFU of CHIKV 181/25 or AF15561 in the left footpad. (**A**) Representative flow cytometry plots and percentages of (**B**) CD11b$^+$Ly-6C$^{hi}$ monocytes or (**C**) CD11b$^{hi}$Ly6C$^+$Ly6G$^+$ neutrophils in the blood. (**D**) Representative flow cytometry plots and numbers of (**E**) CD11b$^+$Ly6C$^{hi}$ monocytes or (**F**) CD11b$^{hi}$Ly6C$^+$Ly6G$^+$ neutrophils in the dLN. Data are combined from two independent experiments (n = 3 (mock) to 7 per group). At 5 dpi, the dLN was analyzed for (**G**) total cells, (**H**) CD19$^+$B220$^+$ B cells, (**I**) TCRβ$^+$CD4$^+$ T cells, and (**J**) TCRβ$^+$CD8$^+$ T cells. Data are combined from 3 independent experiments (n = 4 (mock) or 11 per group). (**K**) Frozen dLN sections were stained for ERTR7$^+$ stromal cells (red), B220$^+$ B cells (blue), or CD8$^+$ T cells (green). Errors bars represent mean ± SEM. Data in (**G-J**) are

combined from 3 independent experiments. Data in (**K**) are representative of 2 independent experiments with 4–5 LNs per group. Statistical significance was determined by one-way ANOVA with Tukey's post-test (*, $P < 0.05$; **, $P < 0.01$; ***, $P < 0.001$).

pathogenic CHIKV infection results in both impaired lymphocyte accumulation and altered lymphocyte localization.

To evaluate which specific Gr-1$^+$ cell type, monocytes or neutrophils, was responsible for the defect in lymphocyte accumulation in the dLN following pathogenic CHIKV infection, we treated mice with an anti-Ly6G mAb to deplete neutrophils alone or an anti-CCR2 mAb [28] to deplete Ly6C$^{hi}$ monocytes alone. In each case, either neutrophils (**Fig 3A and 3B**) or monocytes (**Fig 3G and 3H**) were specifically depleted from the circulation and the dLN. As we did not observe restoration of dLN total cell numbers at 5 dpi in mice depleted of either neutrophils (**Fig 3A–3F**) or Ly6C$^{hi}$ monocytes (**Fig 3G–3L**), the influx of either cell type likely can prevent lymphocyte accumulation in the dLN.

### Early monocyte and neutrophil influx into the dLN inhibits subsequent GC formation and neutralizing Ab responses

In comparison with attenuated CHIKV infection, pathogenic CHIKV infection induces poor GC formation in the dLN and a diminished neutralizing serum Ab response [14, 23]. We hypothesized that the enhanced lymphocyte accumulation and organization in the dLN of mice depleted of monocytes and neutrophils prior to pathogenic infection would improve GC formation. To investigate this idea, mice were treated with a single dose of anti-Gr-1 or isotype control mAb one day prior to pathogenic CHIKV infection, and GC responses in the dLN were evaluated at 14 dpi by staining sections with the GC B cell marker GL7. Anti-Gr-1 mAb-treated mice had increased numbers of GCs per dLN, and the GCs were larger than those of the isotype control group (**Fig 4A–4C**), resulting in an increased total GC area in the dLN (**Fig 4D**). Flow cytometry analysis also showed an increased percentage and number of CD95$^+$GL7$^+$ GC B cells in the dLN of mice infected with pathogenic CHIKV and treated with anti-Gr-1 mAb (**Fig 4E–4G**). Early depletion of monocytes and neutrophils also resulted in greater numbers of CD138$^+$IgD$^{lo}$ plasma cells (PCs) (**Fig 4H–4J**) and increased the number of CHIKV-specific Ab-secreting cells (ASCs) in the dLN (**Fig 4K**). The effects of anti-Gr-1 mAb treatment were most pronounced in the dLN, as we observed modest effects on the percentages of GC B cells and plasma cells in the left inguinal LN and the spleen (**S4 Fig**). Anti-Gr-1 mAb treatment had no effect on the amount of total CHIKV-specific IgG present in serum at day 7, 14, and 28 dpi (**Fig 4L**). However, consistent with the effects of anti-Gr-1 mAb treatment on GC formation, we observed an increase in CHIKV neutralization activity in the serum of anti-Gr-1-treated mice compared with the isotype control group at 28 dpi (**Fig 4M**). Together, these data demonstrate that early depletion of monocytes and neutrophils during pathogenic CHIKV infection improves the endogenous, polyclonal virus-specific GC B cell responses in the dLN and, despite predominantly impacting the dLN, leads to a modest but detectable improvement in serum Ab neutralization activity.

### Mechanisms of pathogenic CHIKV-induced monocyte and neutrophil recruitment

We next investigated the molecular pathways by which monocytes and neutrophils are recruited into the dLN following pathogenic CHIKV infection. Recruitment of Ly6C$^{hi}$ monocytes to inflamed tissue or the dLN in response to viral infection is reportedly CCR2- and type I IFN-dependent [5, 29]. Accordingly, we first investigated the role of type I IFN signaling in

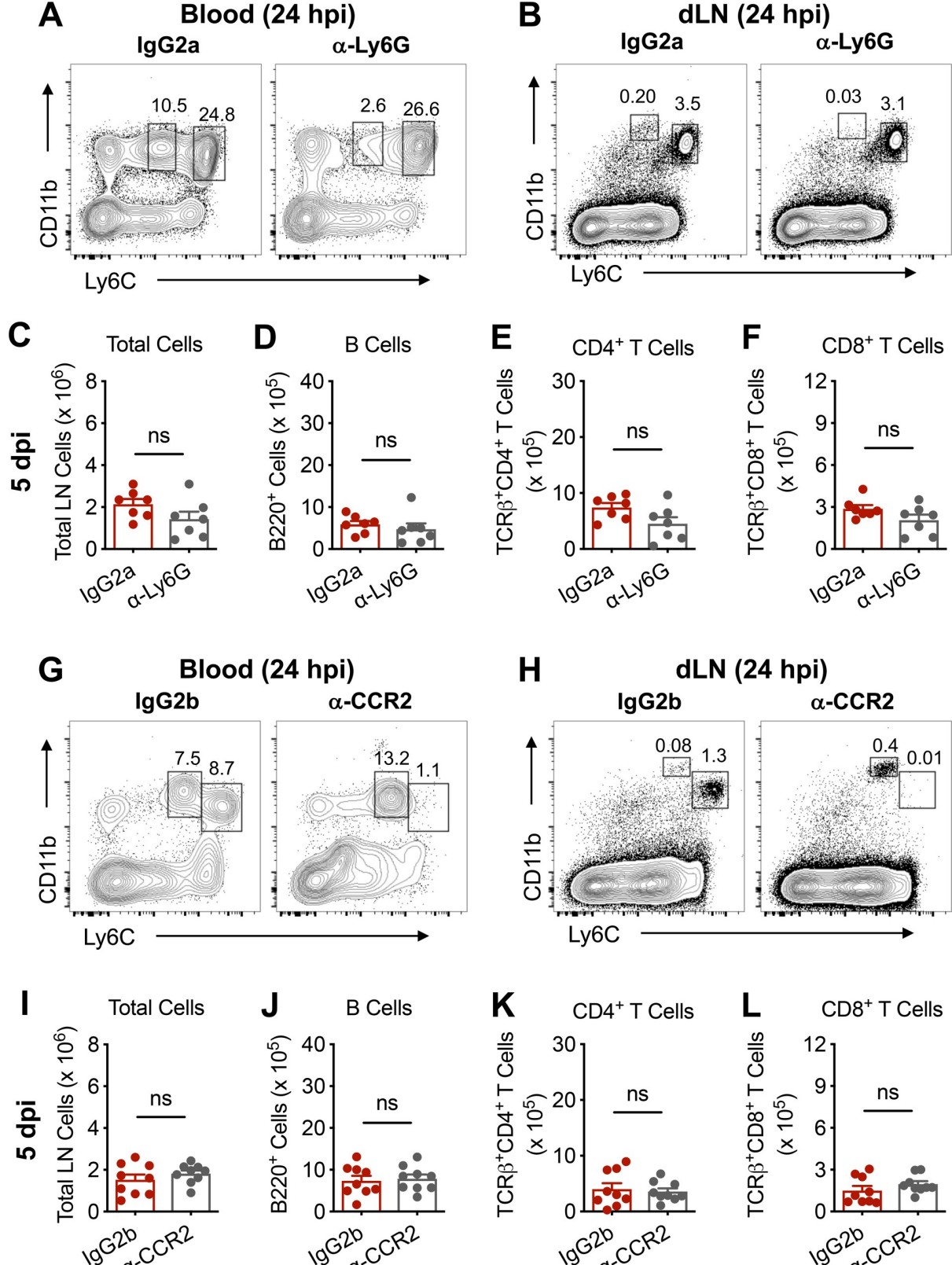

**Fig 3. Monocytes or neutrophils prevent lymphocyte accumulation in the dLN during pathogenic CHIKV infection.** (**A-F**) C57BL/6 mice were treated with 300 µg of IgG2a isotype control Ab or anti-Ly6G mAb via intraperitoneal injection one day prior to inoculation with $10^3$

PFU of CHIKV AF15561 in the left footpad. Representative flow cytometry plots of CD11b$^+$Ly6C$^{hi}$ monocytes or CD11b$^{hi}$Ly6C$^+$Ly6G$^+$ neutrophils in the (**A**) blood or (**B**) dLN. At 5 dpi, the dLN was analyzed for (**C**) total cells, (**D**) CD19$^+$B220$^+$ B cells, (**E**) TCRβ$^+$CD4$^+$ T cells, and (**F**) TCRβ$^+$CD8$^+$ T cells. (**G-L**) C57BL/6 mice were treated with 50 μg of IgG2b isotype control mAb or anti-CCR2 mAb via intraperitoneal injection one day prior to inoculation with 10$^3$ PFU of CHIKV AF15561 in the left footpad. Representative flow cytometry plots of CD11b$^+$Ly6C$^{hi}$ monocytes or CD11b$^{hi}$Ly6C$^+$Ly6G$^+$ neutrophils in the (**G**) blood or (**H**) dLN. At 5 dpi, the dLN was analyzed for (**I**) total cells, (**J**) CD19$^+$B220$^+$ B cells, (**K**) TCRβ$^+$CD4$^+$ T cells, and (**L**) TCRβ$^+$CD8$^+$ T cells. Errors bars represent mean ± SEM. Data are combined from two independent experiments (n = 6–9 per group). Statistical significance was determined by Student's t-test (n.s., not significant).

the accumulation of monocytes and neutrophils in the dLN following pathogenic CHIKV infection. As shown in **Fig 5A**, the total number of dLN cells following pathogenic CHIKV infection was reduced in *Ifnar1$^{-/-}$* compared with wild-type (WT) mice; type I IFN signaling normally inhibits lymphocyte egress [30], but this does not occur in mice lacking the type I IFN receptor and likely contributes to the reduced cell numbers. The percentage of neutrophils was increased in the dLN of *Ifnar1$^{-/-}$* mice compared with WT mice after pathogenic CHIKV infection (**Fig 5B and 5C**), which resulted in a similar total number of neutrophils in the dLN of WT and *Ifnar1$^{-/-}$* mice (**Fig 5D**) despite the overall decrease in total number of dLN cells. The frequencies of Ly6C$^{hi}$ monocytes were similar in the dLN of CHIKV-infected WT and *Ifnar1$^{-/-}$* mice (**Fig 5E**) although the total number of monocytes in the dLN was reduced in *Ifnar1$^{-/-}$* mice.

Because TLR ligands can promote monocyte egress from the bone marrow to sites of infection [31], we evaluated neutrophil and monocyte recruitment to the dLN of mice deficient in MyD88, the canonical adaptor for many TLRs and the interleukin-1 receptor (IL-1R) family [32]. Genetic deletion of MyD88 had minimal impact on the total number of cells in the dLN at 24 hpi (**Fig 5G**). However, MyD88-dependent signals were required for and contributed to neutrophil (**Fig 5H–5J**) and monocyte (**Fig 5H, 5K and 5L**) recruitment, respectively, to the dLN following pathogenic CHIKV infection. Mice treated with anti-IL-1R blocking mAb at the time of infection showed no change in total cells in the dLN compared with control IgG-treated mice (**Fig 5M**), but had reduced neutrophils (percentage and total number; **Fig 5N– 5P**), indicating that the MyD88-IL-1R signaling axis regulates the rapid accumulation of neutrophils in the dLN during pathogenic CHIKV infection. In contrast, the percentage and total number of monocytes were unchanged in the anti-IL-1R-treated group compared with the IgG-treated group (**Fig 5N, 5Q and 5R**). Together, these data demonstrate that in response to pathogenic CHIKV infection 1) type I IFN signaling inhibits and IL-1R signaling promotes, neutrophil recruitment to the dLN, and 2) IL-1R-independent, MyD88-dependent signal(s) promote the recruitment of monocytes to the dLN.

## Nos2 and Nox2 contribute to reduced lymphocyte numbers and disorganization in the dLN

Myeloid cells can suppress immune responses by a variety of mechanisms, including through the production of nitric oxide (NO) via the action of inducible nitric oxide synthase (iNOS, encoded by *Nos2*) [33]. Moreover, superoxide derived from the activity of the phagocyte NADPH oxidase (NOX2, encoded by *gp91phox*, also known as *Cybb*) can combine with NO to form the reactive nitrogen species peroxynitrite (ONOO$^-$), which also can suppress immune responses [34, 35]. Based on this knowledge, we hypothesized that monocytes and neutrophils infiltrating the dLN disrupt lymphocyte accumulation and tissue organization through the actions of iNOS and NOX2. Consistent with the lack of effect of monocyte and neutrophil depletion on accumulation of lymphocytes in the dLN following attenuated CHIKV infection (**S3 Fig**), total cell numbers in the dLN were unchanged in *Nos2$^{-/-}$* and *Nox2$^{-/-}$* mice infected

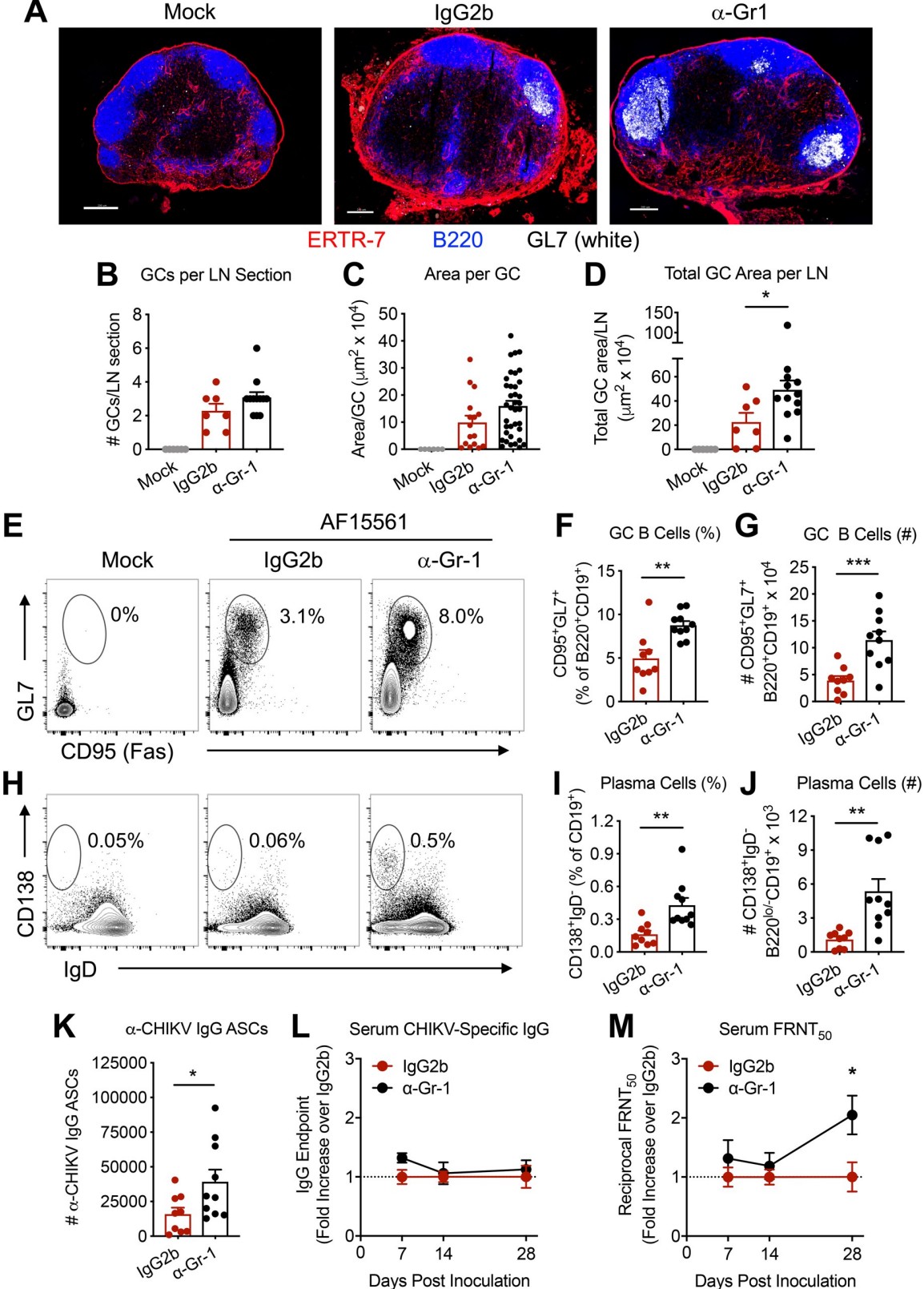

**Fig 4. Depletion of myeloid cells improves dLN B cell responses.** C57BL/6 mice were treated with 300–500 μg IgG2b isotype control mAb or anti-Gr-1 mAb via intraperitoneal injection one day prior to inoculation with $10^3$ PFU of CHIKV AF15561 in the left footpad.

(**A**) At 14 dpi, dLN sections were stained for ERTR7+ stromal cells, B220+ B cells, and GL7+ GC B cells. (**B**) Area per GC, (**C**) Number of GCs per LN section, and (**D**) total GC area per LN were determined. (**E**) Representative flow cytometry plots of GL7+CD95+ GC B cells (gated on CD19+B220+ cells), (**F**) percentage and (**G**) total number of GC B cells in the dLN at 14 dpi. (**H**) Representative flow cytometry plots of CD138+IgD- plasma cells (gated on CD19+), (**I**) percentage and (**J**) total number of plasma cells in the dLN at 14 dpi. (**K**) Number of CHIKV-specific IgG+ antibody secreting cells (ASCs) in the dLN at 14 dpi. Serum collected at the indicated timepoints was assayed for (**L**) total CHIKV-specific IgG by ELISA (expressed as fold change in IgG endpoint over the IgG2b group) and (**M**) neutralizing activity of CHIKV by focus reduction neutralization test (FRNT, expressed as fold change in FRNT$_{50}$ over the IgG2b group). Errors bars represent mean ± SEM. Data in (**A-D**) are derived from 5–6 LNs per group with 7–12 LN sections analyzed per group. Data in (**E-M**) are derived from 2–3 independent experiments with 7–10 mice per group. Statistical significance was determined by Student's t-test in **B-D** (note that significance is only displayed for comparison of IgG2b and α-Gr-1 groups), **F-G**, **I-J**, and **K**; or by two-way ANOVA with Bonferroni's post-test in **L-M** (*, $P < 0.05$; **, $P < 0.01$; ***, $P < 0.001$).

with attenuated CHIKV (Fig 6A). However, total cell numbers in the dLN were partially restored in *Nos2*-/- and *Nox2*-/- mice inoculated with pathogenic CHIKV compared with WT mice (Fig 6A), suggesting that NO and superoxide contribute to reduced lymphocyte numbers in the dLN during pathogenic CHIKV infection. In addition to partial restoration of lymphocyte numbers, the dLNs of *Nos2*-/- and *Nox2*-/- mice infected with pathogenic CHIKV showed improved follicular and paracortical organization, and a more clearly defined B-T cell border compared with the dLN of WT mice (Fig 6B). These data suggest that both Nos2 and Nox2 contribute to the disruption of lymphocyte organization that occurs during pathogenic CHIKV infection.

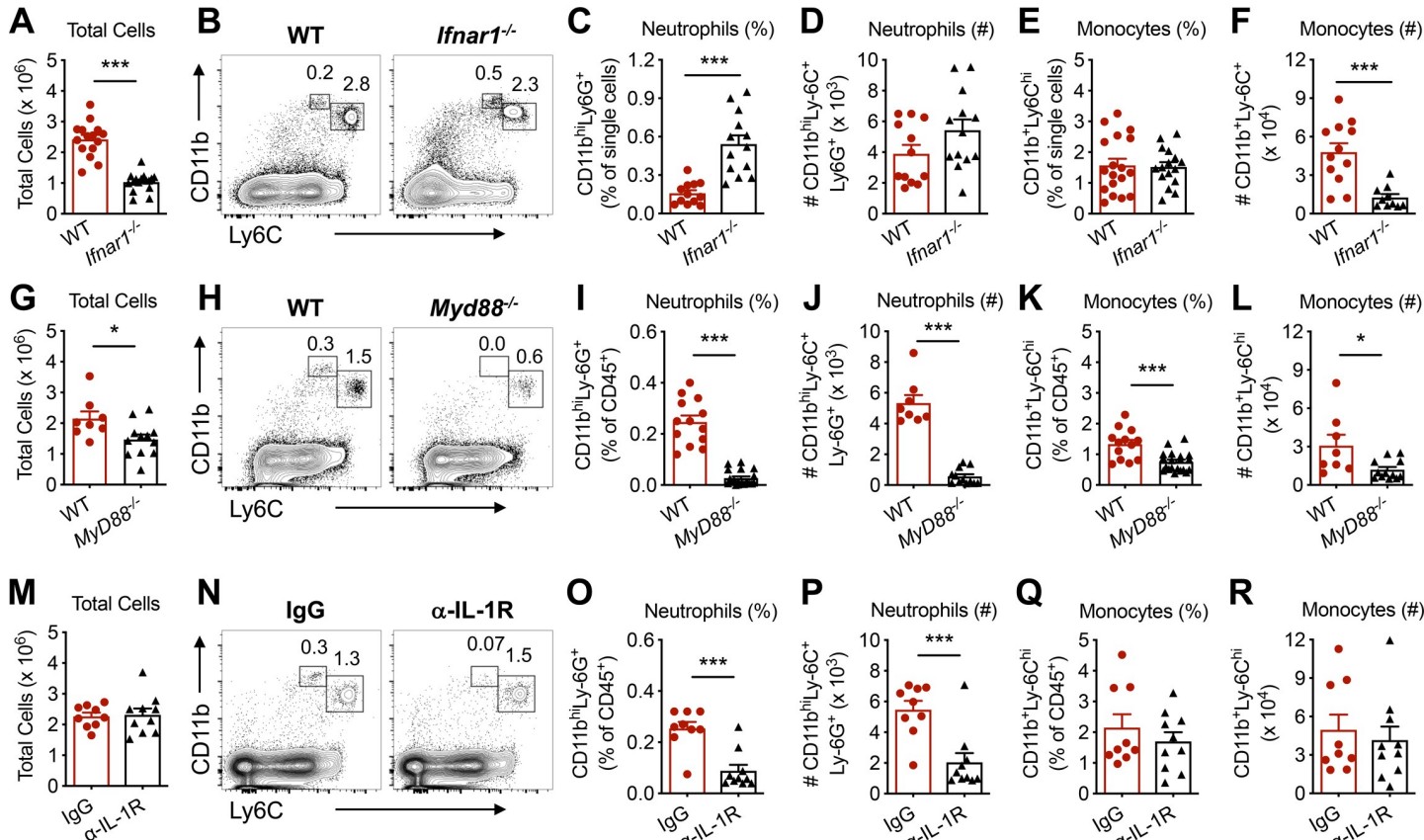

**Fig 5. Mechanisms of pathogenic CHIKV-induced myeloid cell recruitment to the dLN.** Mice were inoculated with 10$^3$ PFU of CHIKV AF15561 in the left footpad. At 24 hpi, total cells, CD11b$^{hi}$Ly6C$^+$Ly6G$^+$ neutrophils, and CD11b$^+$Ly6C$^{hi}$ monocytes in the dLN were enumerated in (**A-F**) WT or *Ifnar1*-/- mice, (**G-L**) WT or *Myd88*-/- mice, or (**M-R**) WT mice treated with 200 μg IgG or α-IL-1R Ab at the time of infection. Errors bars represent mean ± SEM. Data are combined from 2–4 independent experiments (n = 7–16 mice per group). Statistical significance was determined by Student's t-test (*, $P < 0.05$; ***, $P < 0.001$).

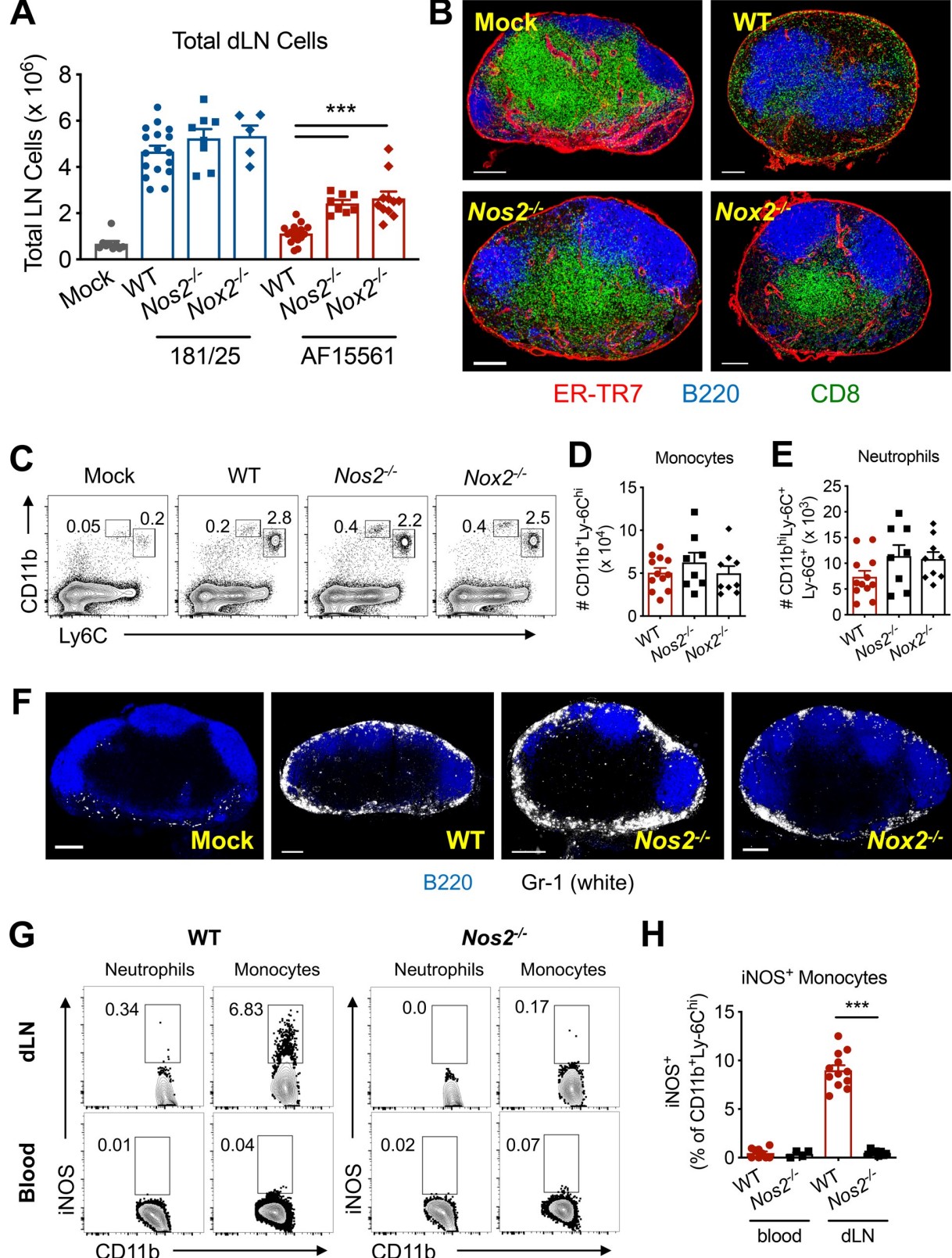

**Fig 6. Nos2 and Nox2 contribute to disrupted dLN architecture and lymphocyte depletion.** WT, *Nos2*[-/-], or *Nox2*[-/-] (*gp91phox*[-/-]) mice were inoculated with PBS or $10^3$ PFU of CHIKV AF15561 in the left footpad. (**A**) At 5 dpi, total cells in the dLN were enumerated. Data are

combined from 3–4 independent experiments (n = 4–16 mice per group). (**B**) Frozen dLN sections (6–7 LNs per group) were stained for ERTR7$^+$ stromal cells (red), B220$^+$ B cells (blue), or CD8$^+$ T cells (green). (**C**) At 24 hpi, representative flow cytometry plots and numbers of (**D**) CD11b$^+$Ly6C$^{hi}$ monocytes or (**E**) CD11b$^{hi}$Ly6C$^+$Ly6G$^+$ neutrophils in the dLN. (**F**) Frozen dLN sections were stained for B220$^+$ B cells (blue), or Gr-1$^+$ monocytes and neutrophils (white). (**G**) Representative flow cytometry plots and (**H**) numbers of iNOS$^+$ cells among circulating (bottom) or dLN (top) CD11b$^+$Ly6C$^{hi}$ monocytes in WT and *Nos2$^{-/-}$* mice. Errors bars represent mean ± SEM. Data are combined from (**A, C-E, G-H**) or representative of (**B, F**) 2–3 independent experiments (n = 4–12 mice per group). Statistical significance in (**A**–displayed only for comparison between AF15561-infected groups) and (**H**) was determined by Student's t-test to compare groups (\*\*\*, $P < 0.001$).

The partial restoration of cell numbers and markedly improved lymphocyte organization in the dLN were not due to a difference in early monocyte and neutrophil infiltration or localization at 24 hpi between WT, *Nos2$^{-/-}$*, and *Nox2$^{-/-}$* mice (**Fig 6C–6F**). To define the iNOS-expressing cell type(s) that mediate the *Nos2*-dependent effects on dLN cell numbers and organization, we analyzed cell populations in the dLN at 24 hpi by flow cytometry. Monocytes elicited by pathogenic CHIKV infection expressed iNOS in the dLN, but not in the blood (**Fig 6G and 6H**). iNOS staining was absent in neutrophils in the dLN (**Fig 6G**), and was absent in *Nos2$^{-/-}$* mice, as expected (**Fig 6G**). Together, these data indicate that expression of Nos2 and Nox2 contribute to impaired lymphocyte accumulation and dLN disorganization, and that monocytes upregulate iNOS expression following entry into the dLN.

## iNOS expression in monocytes is driven by an IFNAR1- and IRF5-dependent pathway

We next evaluated the molecular pathways that promote iNOS expression in monocytes infiltrating the dLN. Many inflammatory stimuli can induce iNOS expression, including type I IFNs [36]. Indeed, type I IFN receptor signaling was required for expression of iNOS in dLN monocytes following CHIKV infection (**Fig 7A and 7B**). Type I IFN signaling and either IRF3 or IRF7 are required for protection from fatal CHIKV infection, with IRF7 responsible for systemic type I IFN production [37, 38]. Unexpectedly, both IRF3 and IRF7 were dispensable for iNOS induction in dLN monocytes (**Fig 7A and 7B**). Instead, *Irf3$^{-/-}$Irf5$^{-/-}$Irf7$^{-/-}$* triple knockout mice showed diminished monocyte iNOS expression similar to that of *Ifnar1$^{-/-}$* mice, suggesting a requirement for IRF5. The dependency on IRF5 for monocyte iNOS expression was confirmed in *Irf5$^{-/-}$* mice (**Fig 7C and 7D**). Together, these data suggest that IRF5- and IFNAR1-dependent signals act locally in the dLN to induce iNOS expression in infiltrating monocytes during pathogenic CHIKV infection.

## iNOS expression in monocytes requires MyD88 and is partially dependent on TLR7

We next assessed the upstream signals required for monocyte expression of iNOS. IRF5 can be activated in a MyD88-dependent manner by TLR7 or TLR9 agonists [39–41]. In the context of some inflammatory stimuli, TLR3-mediated TRIF signaling can synergize with MyD88 for full IRF5 activation [42, 43]. However, pathogenic CHIKV-induced iNOS expression in dLN monocytes was independent of TRIF (**S5A and S5B Fig**). Instead, iNOS expression in infiltrating monocytes was completely dependent on MyD88 (**Fig 7E and 7F**), and partially dependent on TLR7 (**Fig 7E and 7F**), suggesting that multiple MyD88-dependent signals contribute to IRF5 activation. To determine whether the other MyD88-dependent signal responsible for iNOS induction originated from IL-1R, we treated mice with an IL-1R blocking mAb at the time of infection. IL-1R blockade did not alter the percentage of iNOS-expressing monocytes in the dLN (**Fig 7G and 7H**). Together, these data suggest that TLR7 and one or more

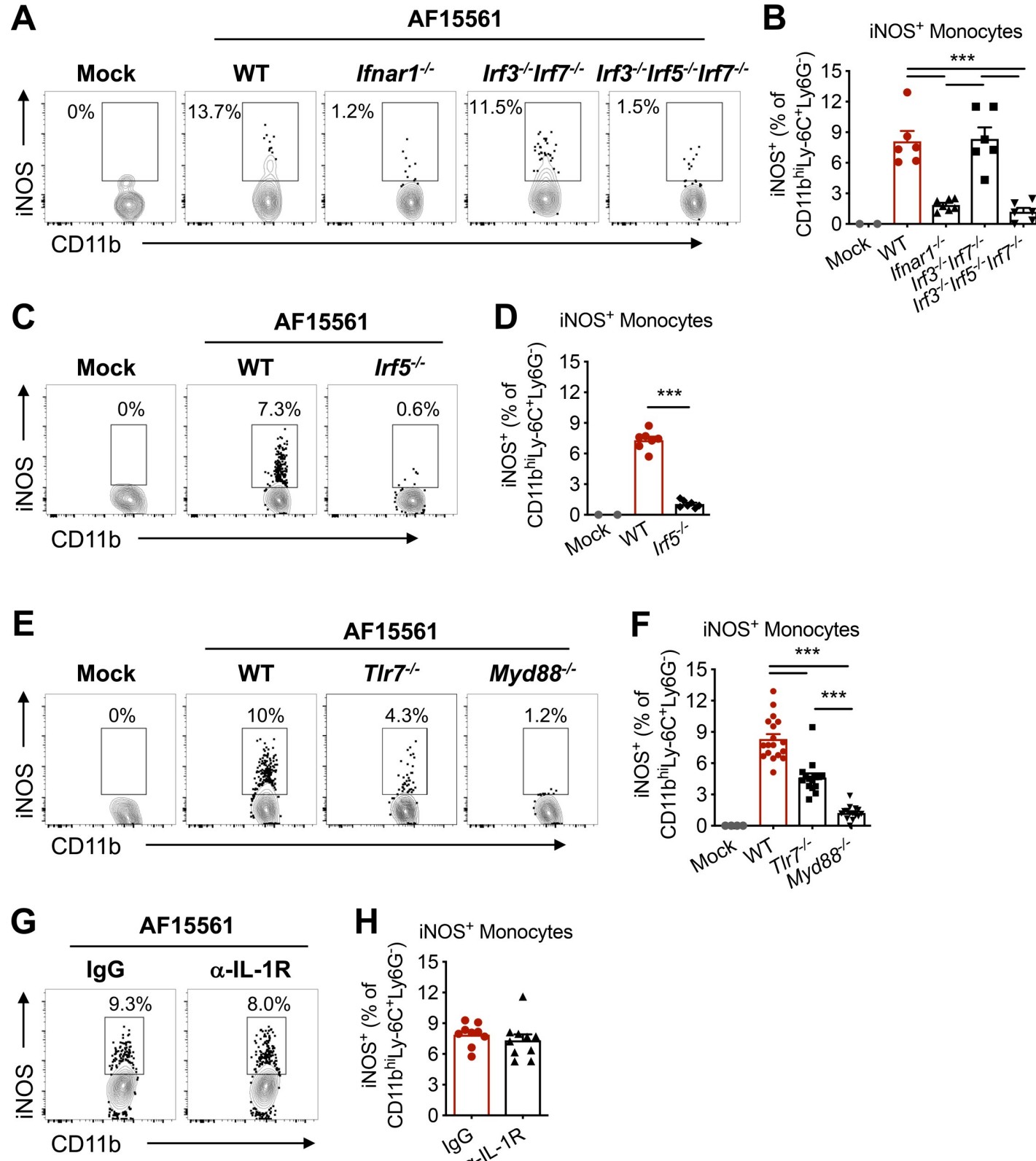

**Fig 7. Mechanisms of iNOS expression in monocytes recruited to the dLN.** Mice were inoculated with PBS (mock) or $10^3$ PFU of CHIKV AF15561 in the left footpad, and the dLN was analyzed at 24 hpi. (**A**) Representative flow cytometry plots and (**B**) percentage of CD11b$^+$Ly6C$^{hi}$ monocytes expressing iNOS in WT, *Ifnar1$^{-/-}$*, *Irf3$^{-/-}$ Irf7$^{-/-}$*, or *Irf3$^{-/-}$ Irf5$^{-/-}$ Irf7$^{-/-}$* mice. (**C**) Representative flow cytometry plots and (**D**) percentage of CD11b$^+$Ly6C$^{hi}$ monocytes expressing iNOS in WT or

*Irf5*<sup>-/-</sup> mice. (**E**) Representative flow cytometry plots and (**F**) percentage of CD11b<sup>+</sup>Ly6C<sup>hi</sup> monocytes expressing iNOS in WT, *Tlr7*<sup>-/-</sup>, or *Myd88*<sup>-/-</sup> mice. (**G-H**) WT mice were treated with 200 μg of IgG or α-IL-1R Ab at the time of infection. (**G**) Representative flow cytometry plots and (**H**) percentage of CD11b<sup>+</sup>Ly6C<sup>hi</sup> monocytes expressing iNOS. Errors bars represent mean ± SEM. Data are combined from 2–3 independent experiments. Statistical significance is displayed only for comparison between infected groups (mock not included) and was determined by one-way ANOVA with Tukey's post-test (**B, F**) or Student's t-test (**D, H**) (***, $P < 0.001$).

additional MyD88-dependent TLRs activate IRF5 to drive iNOS expression in dLN infiltrating monocytes following pathogenic CHIKV infection.

## An IRF5-dependent pathway is activated locally in the dLN

The upregulation of iNOS in monocytes in the dLN, but not in circulation, and the requirement for IFNAR1 and IRF5 in this induction suggested the presence of an IRF5-dependent type I IFN pathway acting locally in the dLN. To investigate this idea, we infected WT, *Irf3*<sup>-/-</sup>*Irf7*<sup>-/-</sup>, and *Irf3*<sup>-/-</sup>*Irf5*<sup>-/-</sup>*Irf7*<sup>-/-</sup> mice with pathogenic CHIKV. At 24 hpi, we measured the induction of interferon stimulated genes (ISGs) including *Ccl2*, *Cxcl10*, *Ifng*, *Il6*, *Ifit1*, and *Rsad2* mRNA [44, 45] in the inoculated foot and in the dLN by quantitative reverse transcription polymerase chain reaction (qRT-PCR). Each of these ISGs was upregulated in both the foot and dLN of CHIKV-infected WT mice at this timepoint (**Fig 8A–8F**). However, upregulation was differentially dependent on IRF3/IRF7 or IRF5 in each tissue. In the inoculated foot, upregulation of all ISGs analyzed was dependent on IRF3/IRF7 (**Fig 8A–8F**). However, in the dLN, the induction of *Ccl2*, *Cxc10*, *Ifng*, and *Il6* mRNA was similar between WT and *Irf3*<sup>-/-</sup>*Irf7*<sup>-/-</sup> mice (**Fig 8A–D**), whereas *Ifit1* and *Rsad2* upregulation in the dLN were only partially dependent on the presence of IRF3/IRF7 (**Fig 8E and 8F**). Instead, for each ISG, the combined deficiency of IRF3, IRF5, and IRF7 resulted in reduced upregulation in the dLN compared with WT or *Irf3*<sup>-/-</sup>*Irf7*<sup>-/-</sup> mice (**Fig 8A–8F**). Together, these data demonstrate that induction of multiple ISGs at the site of inoculation is dependent on IRF3/IRF7, whereas an IRF5-dependent ISG induction pathway is activated in the dLN.

## Discussion

Our results demonstrate that a rapid influx of Ly6C<sup>hi</sup> monocytes and neutrophils into the dLN triggered by infection with pathogenic, persistent strains of CHIKV reduces the accumulation of lymphocytes, disrupts lymphocyte organization, and impairs the development of virus-specific B cell responses. Preventing the early influx of Ly6C<sup>hi</sup> monocytes and neutrophils into the dLN improved accumulation of lymphocytes and follicular organization and enhanced dLN GC formation, plasma cell differentiation, and CHIKV-specific serum neutralizing antibody responses. Lymphocyte accumulation and dLN organization was improved in CHIKV-infected mice lacking Nos2 (iNOS) or the phagocyte NADPH oxidase Nox2. Monocytes, but not other cell types, upregulated iNOS following entry into the dLN in a manner dependent on IFNAR1, IRF5, and MyD88, and partially dependent on TLR7. Although IRF3/IRF7-dependent type I IFN production is required for protection from fatal CHIKV infection [37, 46], we identified an IRF5-dependent pathway acting locally within the dLN following pathogenic CHIKV infection that drives activation of infiltrating monocytes.

Following infection or vaccination, innate immune cells are recruited to lymphoid tissues [5, 47–49]. Myeloid-derived cells can have suppressive effects on adaptive immunity, depending on the local inflammatory milieu. For example, monocytes recruited to the dLN of LCMV-infected mice co-localize with and deplete virus-specific B cells in the interfollicular area [5]. Here, we show that pathogenic, but not acutely cleared CHIKV, similarly induces rapid recruitment of monocytes and neutrophils to the dLN. During pathogenic CHIKV infection, recruited monocytes and neutrophils localized to the SCS and medullary sinuses, with some

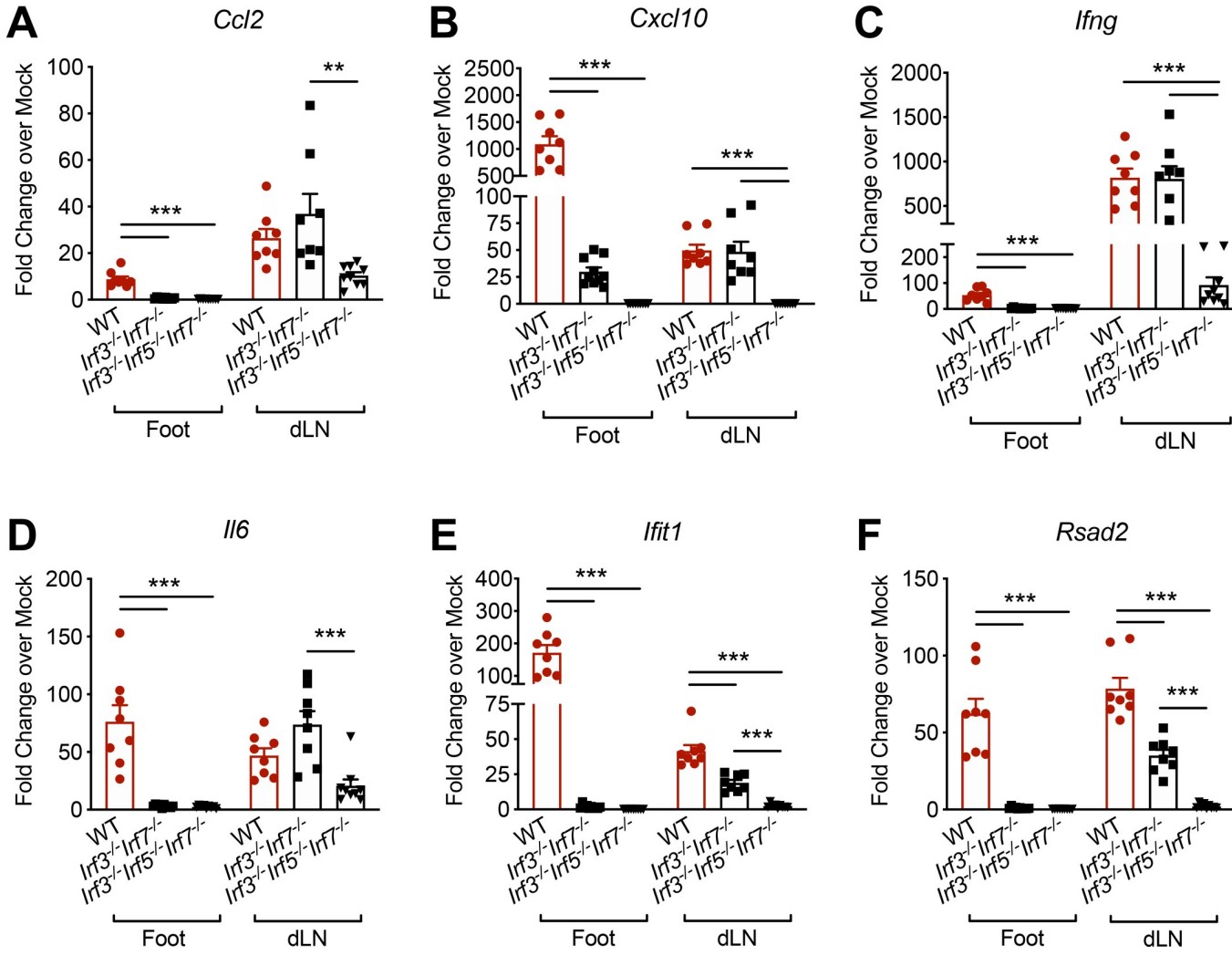

**Fig 8. An IRF5-dependent pathway is activated locally in the dLN.** WT, *Irf3⁻/⁻Irf7⁻/⁻*, or *Irf3⁻/⁻Irf5⁻/⁻ Irf7⁻/⁻* mice were inoculated with PBS (mock) or 10³ PFU of CHIKV AF15561 in the left footpad. At 24 hpi, the inoculated foot and the dLN were collected. qRT-PCR was used to analyze mRNA expression of (A) *Ccl2*, (B) *Cxcl10*, (C) *Ifng*, (D) *Il6*, (E) *Ifit1*, and (F) *Rsad2*. Error bars represent mean ± SEM. Data (8–9 mice per group) are combined from 2 independent experiments. Statistical significance was determined by one-way ANOVA with Tukey's multiple comparisons test. **$P < 0.01$, ***$P < 0.001$.

cells observed within B cell follicles. Ongoing studies are aimed at determining the cell types in the dLN with which monocytes and neutrophils interact, and how their localization determines the subsequent disruption of dLN architecture following infection.

We previously observed that pathogenic CHIKV infection blunts the accumulation of lymphocytes in the dLN by impairing recruitment of naïve lymphocytes from the circulation [23]. Failure to recruit new lymphocytes was due to reduced function of high endothelial venules (HEVs) and decreased production of homeostatic chemokines. Preventing the early influx of monocytes and neutrophils into the dLN restored naïve lymphocyte recruitment, as the lymphocyte numbers in the dLN were uniformly increased. Moreover, preventing the influx of monocytes and neutrophils into the dLN improved organization of B cell follicles, the T cell area, and the B-T cell border. It remains to be determined whether these infiltrating myeloid cells directly also affect the function of HEVs or the production of homeostatic chemokines. Furthermore, monocyte and neutrophil depletion improved virus-specific B cell responses in

the dLN, with more GC B cells and plasma cells generated. This improved GC formation is likely a downstream consequence of boosted lymphocyte numbers and less perturbed dLN organization earlier during infection.

Type I IFNs can promote monocyte recruitment during inflammatory responses [5, 50]. Our data reveal distinct roles for IFNAR1, MyD88, and IL-1R in the recruitment of monocytes and neutrophils to the dLN. Neutrophil recruitment was inhibited by IFNAR1 signaling, but required IL-1R-MyD88 signaling. These findings agree with prior reports demonstrating IL-1R-dependent neutrophil recruitment in response to several inflammatory stimuli [51–53], and a single report showing IFNAR1-mediated inhibition of neutrophil recruitment to inflamed lungs following influenza A virus infection [54]. In contrast to neutrophils, monocyte recruitment to the dLN following pathogenic CHIKV infection was independent of IFNAR1 and IL-1R signaling. However, the percentage and number of monocytes in the dLN was reduced in MyD88-deficient mice, indicating that one or more TLRs likely drive monocyte recruitment.

Total lymphocyte numbers in the dLN were increased in the dLN of $Nos2^{-/-}$ and $Nox2^{-/-}$ mice compared with WT mice following pathogenic CHIKV infection. In addition, and in contrast to WT mice, pathogenic CHIKV infection had minimal effects on the organization of B cell follicles and the T cell zone in the dLN of $Nos2^{-/-}$ and $Nox2^{-/-}$ mice. Flow cytometric analysis revealed that monocytes, but not neutrophils, upregulated iNOS expression following entry into the dLN. During LCMV infection of mice, monocytes infiltrating the dLN induced apoptosis in virus-specific and bystander B cells in a Nos2-dependent manner [5]. Studies with LCMV also observed B cell depletion in lymphoid tissue, however, individual roles for Nos2 [55] or Gr-1+ cells [56] were not defined. In contrast to LCMV infection, extensive lymphocyte death or an increase in apoptotic lymphocytes does not occur in the dLN following pathogenic CHIKV infection [23], suggesting that the infiltrating monocytes do not impair B cell responses by inducing death of lymphocytes. Additionally, lymphocyte numbers were restored only partially in mice deficient for Nos2 or Nox2, suggesting that the combined action of Nos2 and Nox2 or other monocyte and/or neutrophil-derived factors contribute to the impairment of naive lymphocyte accumulation and disruption of LN organization. Depletion of either neutrophils or monocytes alone did not restore total lymphocyte numbers in the dLN, indicating that influx of either cell type is sufficient to prevent accumulation of naïve lymphocytes. Although the improved dLN organization and lymphocyte accumulation observed in Nos2-deficient mice could be ascribed to lack of iNOS expression uniquely in monocytes, the mechanism(s) by which neutrophils impair lymphocyte accumulation and disrupt dLN architecture remain to be determined.

Type I IFN signaling can modulate monocyte activation and differentiation, with context-dependent pro-inflammatory or anti-inflammatory effects [36, 54, 57, 58]. Although not required for recruitment of monocytes to the dLN after pathogenic CHIKV infection, type I IFN signaling drives iNOS expression in monocytes following entry into the dLN. Furthermore, IRF5, but not IRF3 or IRF7, was responsible for induction of iNOS. IRF3 or IRF7 are required for protection from fatal CHIKV infection, with systemic type I IFN production being driven by the IRF7 response [37, 38]. However, the contribution of type I IFN signaling and additional IRFs in other aspects of alphavirus pathogenesis remains poorly defined. Our work suggests that an IRF5-dependent pathway acts locally in draining lymphoid tissue following infection with CHIKV, whereas IRF3/7-dependent pathways are activated at the site of inoculation. Studies in IRF5-overexpressing cells found that IRF5 induces a distinct subset of IFNα genes compared with other IRFs, such as IRF7 [59]. In addition to IFNα, IRF5 also promotes the transcription of genes encoding proinflammatory cytokines, such as *Il12b*, *Tnfa*, and *Il6* [41] and several chemokines [60]. Similar to our observations, IRF5 promotes the

induction of several proinflammatory cytokines and chemokines, as well as recruitment and activation of lymphocytes in the dLN following West Nile Virus (WNV) infection of mice [61]. However, these IRF5-dependent responses were ultimately protective by limiting viral spread and promoting optimal B cell immunity. In contrast, our data suggest that pathogenic strains of CHIKV trigger an IRF5-dependent inflammatory pathway in the dLN that activates infiltrating monocytes and neutrophils, ultimately causing disruption of dLN architecture and decreased virus-specific B cell responses.

IRF5 is expressed constitutively in pDCs, classical or conventional DCs (cDCs), M1 macrophages, and B cells, with further upregulation following activation [41, 59, 62–65]. Upon activation with TLR ligands, IRF5 expression also can be induced in human monocytes, neutrophils, and macrophages [41, 62, 66]. Thus, many immune cell types can express IRF5, either constitutively or upon activation. Viral infections or agonists of TLR7 or TLR9 can activate IRF5 in DCs *in vitro* in a MyD88-dependent manner [39–41]. Following pathogenic CHIKV infection, we observed a partial role for TLR7 in iNOS upregulation, suggesting that recognition of endosomal viral RNA contributes to monocyte activation. However, our data suggest role(s) for other MyD88-dependent TLR(s) in activating IRF5 to promote iNOS expression in infiltrating monocytes. One or more of these TLRs may be activated by endogenous damage-associated molecular patterns (DAMPs) released from damaged or dying cells during the early stages of CHIKV infection, either at the site of inoculation or in the dLN. The cell-intrinsic requirements for TLR(s), IRF5, and IFNAR1 expression in dLN monocyte iNOS upregulation during CHIKV infection remain to be determined through the use of conditional knockout mice.

In summary, we found that a rapid influx of Ly6C$^{hi}$ monocytes and neutrophils to the dLN impairs lymphocyte accumulation, dLN organization, and downstream antiviral B cell responses to pathogenic CHIKV infection in a manner partially dependent on Nos2 and Nox2. Furthermore, we identified a role for MyD88-dependent IRF5-driven type I IFN signaling in the dLN that promotes iNOS expression in infiltrating monocytes. Our findings reveal an additional mechanism that dampens the development of the anti-CHIKV Ab response during pathogenic CHIKV infection, and underscore how local innate immune responses in draining lymphoid tissue dictate the magnitude of downstream adaptive immunity.

## Materials and methods

### Ethics

This study was conducted in accordance with the recommendations in the Guide for the Care and Use of Laboratory Animals and the AVMA Guidelines for the Euthanasia of Animals. All animal experiments were performed with the approval of the Institutional Animal Care and Use Committee at the University School of Medicine (Assurance Number: A3269-01) or at the Washington University School of Medicine (Assurance Number A3381-01). Experimental animals were humanely euthanized at defined endpoints by exposure to isoflurane vapors followed by thoracotomy.

### Mouse experiments

WT C57BL/6 (000664), *Nos2*$^{-/-}$ (002609; B6.129P2-Nos2$^{tm1Lau}$), *gp91*$^{phox-/-}$ (002365; B6.129S6-Cybb$^{tm1Din}$), *Tlr7*$^{-/-}$ (008380; B6.129S1-Tlr7$^{tm1Flv}$), *Myd88*$^{-/-}$ (009088; B6.129P2(SJL)-Myd88$^{tm1.1Defr}$), and *LT*$^{-/-}$ (002258; B6.129S2-Lta$^{tm1Dch}$/J) mice were obtained from The Jackson Laboratory. *Irf5*$^{-/-}$ mice were a gift of T. Taniguchi (Tokyo, Japan), obtained from I. Rifkin (Boston, MA), and were backcrossed for eight generations. After detection of a homozygous Dock2 mutation in this line, it was backcrossed for five additional generations to obtain *Irf5*$^{-/-}$

*Dock2^{wt/wt}* mice [61]. *Irf3^{-/-}Irf5^{-/-}Irf7^{-/-}* mice were a gift from Dr. Sujan Shresta (La Jolla Institute for Immunology) and originally generated at Washington University [45]. *Nos2^{-/-}*, *gp91^{phox-/-}* (*Nox2^{-/-}*), *Tlr7^{-/-}*, *Myd88^{-/-}*, *Irf5^{-/-}*, *Irf3^{-/-}Irf7^{-/-}*, *Irf3^{-/-}Irf5^{-/-}Irf7^{-/-}*, and *Ifnar1^{-/-}* mice were bred and housed at the University of Colorado School of Medicine or Washington University under specific pathogen-free conditions. Mice were used at 3–4 weeks of age. Virus inoculations were performed under isoflurane anaesthesia and all efforts were made to minimize animal suffering. All mouse experiments were performed in an animal biosafety level 3 laboratory.

Mice were anesthetized with isoflurane and inoculated with $10^3$ PFU of CHIKV subcutaneously in the left rear footpad in a volume of 10 μL of PBS/1% FBS. Mice were sacrificed after exposure to isoflurane vapors followed by thoracotomy and perfusion with PBS at the indicated time points. In some experiments, mice were treated with 300–500 μg of IgG2b isotype control Ab (LTF-2), anti-Gr-1 mAb (RB6-8C5), IgG2a isotype control Ab (2A3), or anti-Ly-6G mAb (1A8) (all from BioXCell) i.p. the day prior to infection. For IL-1R blockade, mice were treated with 200 μg Armenian hamster IgG control Ab or anti-IL-1R Ab (JAMA-147) (both from BioXCell) at the time of infection.

## Cell lines and viruses

BHK-21 cells (American Type Culture Collection) were cultured at 37˚C in Minimum Essential Medium Alpha (MEMα) supplemented with 10% fetal bovine serum (FBS) and 10% tryptose phosphate broth (TPB). Vero cells (American Type Culture Collection) were cultured at 37˚C in Dulbecco's Modified Eagle Medium/Nutrient Mixture F-12 (DMEM/F-12) with 10% FBS.

CHIKV cDNA clones encoding 181/25 and AF15561 have been described previously [67]. Stocks of infectious CHIKV strains were generated from cDNA clones and titered by plaque assay on BHK-21 cells as previously described [68]. The number of particles/mL of each virus stock was determined by qRT-PCR. All experiments were performed under biosafety level 3 conditions.

## Isolation of cells from lymphoid tissue and flow cytometry

The left popliteal LN was gently homogenized in a Biomasher II tissue homogenizer (Kimble-Chase) in RPMI 1640 (HyClone) supplemented with 10% FBS. For flow cytometric analysis of infiltrating footpad or dLN cells, 2.5 mg/ml collagenase type I (Worthington Biochemical) and 17 μg/ml DNase I (Roche) were added to RPMI 1640 with 10% FBS, and samples were incubated for 1 h at 37˚C with shaking (130 rpm). Following erythrocyte lysis (footpad only), cells were passed through a 100 μm cell strainer (BD Falcon) and total viable cells were determined by trypan blue exclusion. For analysis of circulating immune cells, blood was collected from the inferior vena cava into 50 μL heparin. Following erythrocyte lysis, cells were stained as below.

Single cell suspensions were incubated with anti-mouse FcγRIII/II (2.4G2; BD Pharmingen) for 20 min on ice and then stained with antibodies against the following cell surface markers in 1X PBS/2% FBS for 1 h on ice: CD45 (30-F11), CD11b (M1/70), Ly-6C (HK1.4), Ly-6G (1A8), CD19 (6D5), B220 (RA3-6B2), TCRβ (H57-597), CD4 (RM4-5), CD8 (53–6.7), CD95 (Fas; 15A7), GL7 (GL7), IgD (11-26c.2a), and CD138 (281–2) (all Abs from BioLegend, BD Bioscience, or Thermo Fisher). Cells were washed three times in PBS/2% FBS and fixed overnight in 1X PBS/1% PFA prior to analysis. For intracellular analysis of iNOS expression, cells were stained first with antibodies against cell surface markers, fixed for 15 min at RT in PBS/1% paraformaldehyde (PFA)/0.1% saponin, and then stained with anti-iNOS Ab (clone CXNFT; Thermo Fisher) in PBS/2% FBS/0.1% saponin for 1 h at 4˚C. Cells were washed three

times in PBS/2% FBS/0.1% saponin, fixed overnight in 1X PBS/1% PFA, and analyzed on a BD LSR Fortessa cytometer using FACSDiva software. Further analysis was performed using FlowJo software (Tree Star).

## CHIKV ELISA and focus reduction neutralization test (FRNT)

CHIKV-specific antibodies in mouse sera were measured using a virion-based ELISA as described [14]. Endpoint titers were defined as the reciprocal of the last dilution to have an absorbance two times greater than background. Blank wells receiving no serum were used to quantify background signal. Fold change was calculated by dividing the IgG endpoint of each sample by the average IgG endpoint of the group receiving IgG2b control mAb.

For FRNT assays, Vero cells were seeded in 96-well plates. Serum samples were heat-inactivated and serially diluted in DMEM/F12 medium with 2% FBS in 96-well plates. Approximately 100 focus-forming units (FFU) of virus stock was added to each well and the serum plus virus mixture was incubated for 1 h at 37°C. At the end of 1 h, medium was removed from Vero cells and serum sample + virus mixture was added for 2 h at 37°C. After 2 h, sample was removed and cells were overlaid with 0.5% methylcellulose in MEM/5% FBS and incubated 18 h at 37°C. Cells were fixed with 1% PFA and probed with 500 ng/mL of the anti-CHK-11 mAb [12] diluted in 1X PBS/0.1% saponin/0.1% bovine serum albumin (BSA) for 2 h at RT. After washing, cells were incubated with horseradish peroxidase (HRP)-conjugated goat anti-mouse IgG (Southern Biotech, 1:2000) for 1.5–2 h at RT. After washing, CHIKV-positive foci were visualized with TrueBlue substrate (Fisher) and counted using a CTL Biospot analyzer and Biospot software (Cellular Technology). Percent infectivity was calculated compared to a virus only (no serum) control. The $FRNT_{50}$ value was defined as the reciprocal of the last dilution to exhibit 50% infectivity. Fold change was calculated by dividing the $FRNT_{50}$ of each sample by the average $FRNT_{50}$ of the group receiving IgG2b control Ab.

## CHIKV IgG enzyme linked immunosorbent assay (ELISPOT)

For quantification of CHIKV-specific IgG+ ASCs, dLNs were removed and processed into a single cell suspension as above. Two-fold dilutions of cells (starting at $1 \times 10^5$) were plated on filter plates (EMP Millipore MultiScreen HTS-IP) coated with $10^8$ particles/well of CHIKV AF15561 and incubated for 5 h at 37°C. Biotin-conjugated anti-IgG and streptavidin-conjugated horseradish peroxidase (HRP) (both from Southern Biotech) were used for detection. For development, 200 μL 3-amino-9-ethylcarbazole (AEC) solution (Sigma) was added to 9 ml 0.1 M sodium acetate buffer containing 4 μL 30% $H_2O_2$. Spots were analyzed on a C.T.L. ELISPOT Reader using ImmunoSpot 6.1 software (Cellular Technology).

## Gene expression analysis from the dLN and foot

The dLN or left foot was dissected and homogenized in TRIzol Reagent (Life Technologies) for RNA analysis with a MagNA Lyser (Roche). RNA was isolated using the PureLink RNA Mini kit (Thermo Fisher) with on-column DNase treatment. Gene expression was quantified by RT-qPCR using available Taqman gene expression assays (Thermo Fisher). Expression of each gene was normalized to 18S and expressed as fold change over mock samples.

## Viral RNA analysis

Ankle and spleen tissues were dissected and homogenized in TRIzol Reagent (Life Technologies) for RNA analysis with a MagNA Lyser (Roche). Viral RNA in tissues was quantified by RT-qPCR as previously described [23].

## Immunofluorescence and confocal microscopy

Lymph nodes were fixed in 1 mL of phosphate buffer containing 0.1 M L-lysine, 2% PFA, and 2.1 mg/mL NaIO$_4$ at pH 7.4 for 24 h at 4˚C, followed by incubation in 30% sucrose phosphate-buffered solution for 48 h, then in 30% sucrose/PBS for 24 hr. LNs were then embedded in optimal-cutting-temperature medium (Electron Microscopy Sciences) and frozen in dry-ice-cooled isopentane. Eighteen-μm sections were cut on a Leica cryostat (Leica Microsystems). Sections were blocked with 5% goat, donkey, bovine, rat or rabbit serum and then stained with one or more of the following: B220 (clone RA3-6B2, ThermoFisher), CD8α (clone 53–6.7, ThermoFisher), ERTR-7 (Rat monoclonal, BioXCell), Gr-1 (clone RB6-8C5, BioXCell), and GL7 (clone GL7, BioLegend). Images were acquired using identical photomultiplier tube (PMT) and laser power settings on a Leica SP5 confocal equipped with HyD detectors (Leica). Confocal microscopy images were performed of the entire popliteal lymph node (representing approximately a 7 mm$^2$ imaged area) and individual fields (tiles) were merged into a single image file. Images were analysed using Imaris v9.02 software (Bitplane). The total number of GCs, area per GC, and total GC area were calculated per 18-μM dLN section automatically using Imaris' surfaces function based upon GL7 staining. The total GC number was also confirmed manually.

## Quantification and statistical analysis

Statistical details for each experiment are found in the corresponding figure legends. Statistical analyses were conducted using GraphPad Prism 6.0.

## Supporting information

**S1 Fig. Peripheral lymph organs are required for clearance of CHIKV 181/25 infection in joint-associated tissue.** WT or congenic lymphotoxin alpha-deficient ($LT^{-/-}$) C57BL/6 mice were inoculated with 10$^3$ PFU of CHIKV 181/25 or AF15561 in the left footpad. At 28 dpi, RNA was extracted from the (**A**) right ankle and (**B**) spleen and viral loads were determined by RT-qPCR. Data are from one independent experiment.
(TIF)

**S2 Fig. Circulating myeloid cells elicited by pathogenic CHIKV infection infiltrate the dLN before the site of inoculation.** C57BL/6 mice were inoculated with PBS (mock) or with 10$^3$ PFU of CHIKV 181/25 or AF15561 in the left footpad. At 24 hpi, the blood and left foot were analyzed by flow cytometry. (**A**) Representative flow cytometry plots and percentages of (**B**) CD11b$^+$Ly6C$^{hi}$ monocytes or (**C**) CD11b$^{hi}$Ly6C$^+$Ly6G$^+$ neutrophils in the blood. (**D**) Representative flow cytometry plots and numbers of (**E**) monocytes or (**F**) neutrophils in the left foot. Data are combined from two (**A-C**) or five (**D-F**) independent experiments (n = 4–21 per group). Statistical significance was determined by one-way ANOVA with Tukey's post-test (***, $P < 0.001$).
(TIF)

**S3 Fig. Anti-Gr-1 mAb treatment does not affect lymphocyte accumulation in dLN following attenuated CHIKV infection.** C57BL/6 mice were treated with 500 μg of IgG2b isotype control mAb or anti-Gr-1 mAb via intraperitoneal injection one day prior to inoculation with 10$^3$ PFU of CHIKV 181/25 in the left footpad. At 5 dpi, total cells in the dLN were enumerated. Data are combined from 2 independent experiments.
(TIF)

**S4 Fig. The effects of anti-Gr-1 mAb treatment are largely confined to the dLN.** C57BL/6 mice were treated with 300–500 μg IgG2b isotype control mAb or anti-Gr-1 mAb i.p. one day prior to inoculation with $10^3$ PFU of CHIKV AF15561 in the left footpad. (**A**) Representative flow cytometry plots of GL7$^+$CD95$^+$ GC B cells (gated on CD19$^+$B220$^+$ cells), (**B**) percentage and (**C**) total number of GC B cells in the left inguinal LN at 14 dpi. (**D**) Representative flow cytometry plots of CD138$^+$IgD$^-$ plasma cells (gated on CD19$^+$), (**E**) percentage and (**F**) total number of plasma cells in the left inguinal LN at 14 dpi. (**G**) Representative flow cytometry plots of GL7$^+$CD95$^+$ GC B cells (gated on CD19$^+$B220$^+$ cells), (**H**) percentage and (**I**) total number of GC B cells in the spleen at 14 dpi. (**J**) Representative flow cytometry plots of CD138$^+$IgD$^-$ plasma cells (gated on CD19$^+$), (**K**) percentage and (**L**) total number of plasma cells in the spleen at 14 dpi. Errors bars represent mean ± SEM. Data are derived from 2 independent experiments. Statistical significance was determined by Student's t-test. $^*P < 0.05$, $^{**}P < 0.01$.
(TIF)

**S5 Fig. iNOS expression in monocytes is independent of TRIF.** C57BL/6 mice were inoculated with PBS (mock) or with $10^3$ PFU of CHIKV AF15561 in the left footpad and the dLN was analyzed at 24 hpi. (**A**) Percentage and (**B**) representative flow cytometry plots of CD11b$^+$Ly6C$^{hi}$ monocytes expressing iNOS in WT or *Trif$^{-/-}$* mice. Data are combined from 2 independent experiments.
(TIF)

## Acknowledgments

We thank Jonathan Miner for providing the *Irf5$^{-/-}$* mice.

## Author Contributions

**Conceptualization:** Mary K. McCarthy, Thomas E. Morrison.

**Formal analysis:** Mary K. McCarthy, Heather D. Hickman.

**Funding acquisition:** Mary K. McCarthy, Michael S. Diamond, Heather D. Hickman, Thomas E. Morrison.

**Investigation:** Mary K. McCarthy, Glennys V. Reynoso, Emma S. Winkler.

**Project administration:** Thomas E. Morrison.

**Resources:** Matthias Mack, Michael S. Diamond, Heather D. Hickman.

**Supervision:** Michael S. Diamond, Heather D. Hickman, Thomas E. Morrison.

**Writing – original draft:** Mary K. McCarthy, Thomas E. Morrison.

**Writing – review & editing:** Mary K. McCarthy, Emma S. Winkler, Michael S. Diamond, Heather D. Hickman, Thomas E. Morrison.

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
