## [Decision Letter · Decision Letter 0]

1 Oct 2019

Dear Dr. Morrison,

Thank you very much for submitting your manuscript "MyD88-dependent influx of monocytes and neutrophils impairs lymph node B cell responses to CHIKV infection via Irf5, Nos2 and Nox2" (PPATHOGENS-D-19-01494) for review by PLOS Pathogens. Your manuscript was fully evaluated at the editorial level and by independent peer reviewers. The reviewers appreciated the attention to an important problem, but raised some substantial concerns about the manuscript as it currently stands. These issues must be addressed before we would be willing to consider a revised version of your study. We cannot, of course, promise publication at that time.

*customized message from Marco: Hello Thomas, I appologize for the original delay in sending this work out to review, as it was stalled in my feed while I was on vacation. As you will see from the reviews, each reviewer viewed the work in an overall positive light, but requested and/or suggested further experimentation that will improve the data and their interpretation. I ask then, that you treat this as a 'major' revision, performing additional work where it is possible, and reasonable in terms of time; in which case the revised work should more easily and rapidly move through review during resubmission.*

We therefore ask you to modify the manuscript according to the review recommendations before we can consider your manuscript for acceptance. Your revisions should address the specific points made by each reviewer.

(1) A letter containing a detailed list of your responses to the review comments and a description of the changes you have made in the manuscript. Please note while forming your response, if your article is accepted, you may have the opportunity to make the peer review history publicly available. The record will include editor decision letters (with reviews) and your responses to reviewer comments. If eligible, we will contact you to opt in or out.

(2) Two versions of the manuscript: one with either highlights or tracked changes denoting where the text has been changed; the other a clean version (uploaded as the manuscript file).

Additionally, to enhance the reproducibility of your results, PLOS recommends that you deposit your laboratory protocols in protocols.io, where a protocol can be assigned its own identifier (DOI) such that it can be cited independently in the future. For instructions see http://journals.plos.org/plospathogens/s/submission-guidelines#loc-materials-and-methods

We hope to receive your revised manuscript within 60 days. If you anticipate any delay in its return, we ask that you let us know the expected resubmission date by replying to this email. Revised manuscripts received beyond 60 days may require evaluation and peer review similar to that applied to newly submitted manuscripts.

[LINK]

Sincerely,

Marco Vignuzzi, Ph.D.

Section Editor

PLOS Pathogens

Marco Vignuzzi

Section Editor

PLOS Pathogens

Kasturi Haldar

Editor-in-Chief

PLOS Pathogens

orcid.org/0000-0001-5065-158X

Grant McFadden

Editor-in-Chief

PLOS Pathogens

orcid.org/0000-0002-2556-3526

Reviewer's Responses to Questions

**Part I - Summary**

Reviewer #1: In this study by McCarthy et al., the authors follow up on a published observation that infection with a pathogenic strain of CHIKV leads to an overall disorganization of the dLN including the failure of lymphocytes to accumulate and GCs to develop. Here, the authors perform a detailed, mechanistic analysis of what leads to this phenotype and show nicely that in mice infected with a “pathogenic” strain AF15561 there are increases in monocytes and neutrophils in the dLN early in infection and depletion of these cells leads to an accumulation of lymphocytes and restoration of the dLN organization after CHIKV infection. They go on to show that depletion of Gr-1 positive cells also increases CHIKV specific B-cell responses. To continue their mechanistic studies, they perform a beautiful set of genetic experiments to show that the monocyte recruitment is dependent on MyD88, Nos2 and Nox2 are involved in the disruption of the dLN during CHIKV infection, and iNos expression is controlled by an IFNAR and IRF5 dependent pathway. Finally, they show that there are distinct pathways activation ISGs in the footpad (IRF3/IRF7 dependent) and the dLN (IRF5 dependent). Overall, this study was well-written, easy to follow, and is novel, contributing to multiple areas of CHIKV infection. However, I had a few concerns that will should be addressed.

Reviewer #2: Previously, this group has shown that pathogenic CHIKV strains persist in the joint tissues of WT mice while the attenuated vaccine strain was cleared in a fashion dependent on virus-specific antibody responses. In this study, McCarthy et. al. identified two cell types, neutrophils and inflammatory monocytes, that when depleted increase lymphocyte accumulation and improve draining lymph node (dLN) organization and CHIKV-specific B cell responses prior to infection with a pathogenic and chronic CHIKV strain (AF15561). This is due to iNOS and NOX2 expression in a IRF5/IFNAR1-dependent and partially TLR7-dependent mechanism.

This work is important and novel in two main ways. The authors have 1) defined the cell types and pathways responsible for differential B cell responses to acute versus chronic CHIKV infections, and 2) identified differential innate immune gene expression programs in the site of inoculation versus dLN. This study has left some interesting remaining questions: 1) what are the MyD88-dependent pathways in addition to TLR7, and 2) what is the cell type requirement for IRF5? The manuscript is well-written, and the model is nicely developed and supported by data that demonstrates both rigor and statistical power. Overall, this is a very nice study. Only a few issues need to be addressed.

Reviewer #3: The report from McCarthy et al describes mechanisms behind differences in the control of pathogenic versus attenuated chikungunya virus strains. Multiple transgenic mouse strains are used in this detailed description of the events up- and downstream of the role monocytes and neutrophils play in disrupting effective B-cell responses in draining lymph nodes. This is an excellent, and very thorough study, likely to be of general interest and highly impactful for the field.

The differences between attenuated and pathogenic virus in terms of monocyte recruitment into draining LNs, the mechanisms behind this and the downstream consequences to LNs and germinal center formation are all clearly described and supported. The only criticism of the study is that although there are claims that these local innate immune responses “…..dictate the effectiveness of downstream adaptive immunity”, there is very little actual evidence to support this. There is no discussion as to whether and how these minimal changes may represent the stark differences in pathogenicity seen. More detailed time-courses, isotype differences and/or study of anti-CHIKV memory B-cells or long-lived plasma cells from outside the LN may also identify more significant differences. There is also no discussion of impact on disease outcomes in the mice. Obviously there are issues with downstream effects on viral clearance by these cells in tissues, however with careful timing, or in some of the mouse strains, it seems like some mitigation of disease should be observed if the germinal center disruption really plays a significant role in pathogenesis.

Despite this caveat, the study remains highly informative and will move the field forward significantly.

**Part II – Major Issues: Key Experiments Required for Acceptance**

Reviewer #1: 1. The authors make the point several times that the clearance of attenuated, acute CHIKV infections are due to B-cell and Ab responses (Lines 80-81 for example) and that during persistent, pathogenic CHIKV infections “virus-specific B-cell responses may be impaired” (Lines 82-83). I took this as an important point of the paper and that the authors were hypothesizing that the impaired B cell response is what is responsible or partly responsible for persistence of pathogenic CHIKV. The authors do address B-cell specific responses in Figure 4 and nicely show increases in CHIKV specific B-cells. However, it is unclear if this increase in B cells leads to increased clearance of pathogenic CHIKV and is a potential mechanism for persistence. The authors should measure viral titers in Gr-1 depleted mice to determine if pathogenic CHIKV has increased clearance after monocyte depletion and B cell recruitment. If this was not the point of the paper the authors may want to focus the writing on the disorganized dLN, etc phenotype and leave out viral clearance and persistence until the discussion. If this was addressed but didn’t work, the authors should include some lines in the discussion about potential mechanisms.

2. In the title and their conclusions, the authors make the point that this MyD88-dependent recruitment of monocytes impairs CHIKV B-cell responses. However, the authors only show CHIKV specific B cells after Gr-1 treatment. What about in My88 deficient mice? Nos, Nox, and IRF5 deficient mice? The authors should address lymphocyte recruitment to dLN and CHIKV Abs in these mice.

3. The use of “attenuated” and “pathogenic, persistent” to describe the CHIKV strains was a bit confusing. There are other pathogenic strains of CHIKV that may not be persistent strains (IOL for example) and relative to each other, some strains may be considered “attenuated”. At this point, it is unclear if this same mechanism holds up for all of these strains and until it’s address I suggest calling these viruses as they are “vaccine” and “AF15561” or “parent”.

Reviewer #2: 1. It is not clear what the molecular mechanism and functional consequences of improved B cell responses are. In the previous study (Hawman et al, Cell Rep 2016), the pathogenic CHIKV strain AF15561 has evolved ways to evade neutralizing antibody responses against E2. How does myeloid cell depletion in AF15561-infected mice improve neutralization activity of serum? Has the virus mutated? Is this increase in serum neutralization activity sufficient to restrict viral replication in the joint tissues and facilitate viral clearance? It would be informative to perform passive transfer of serum from Gr-1 antibody treated mice and include more discussions on this data.

2. In Figure 1A, dLN from AF15561-infected mice has significantly more cells in general. What are these cells other than the neutrophils and monocytes? It would be informative to immunophenotype all the cells in the dLN. Since depletion of both neutrophil and monocytes does not completely rescue CD8+ T cells from the dLN (Fig. 2K) and IRF5 is required and constitutively expressed in DCs, it is possible that DCs might play a role in viral persistence. Does pathogenic CHIKV infection increase the influx of DCs into dLN?

Reviewer #3: (No Response)

**Part III – Minor Issues: Editorial and Data Presentation Modifications**

Reviewer #1: 1. Figure 1B: There is a * as significant that is not reflected in the figure legend.

2. Figure 7: The figure legend is swapped with the figure. For example, Figure 7F is described in what is B in the legend. Figure G-H are described in C-D of the legend.

Reviewer #2: 1. In a few occasions, the data does not fully support the model and the writing needs to be modified to take that into consideration. For example, in lines 209-210, the authors stated that “IL-1R-independent, MyD88-dependent signal(s) promote the recruitment of monocytes to the dLN”. However, monocyte number is significantly reduced in the dLN of IFNAR knockout mice although the percentage of monocytes is not affected, suggesting that IFN signaling might have an effect on monocyte accumulation. Additionally, most of the ISGs tested in Figure 8 demonstrate strict IRF5 dependency in dLN except for Ifit1 and Rsad2. Is it possible that the dichotomy of IRF3/7-dependent ISG expression near site of inoculation and IRF5-dependent ISG expression in dLN is gene specific?

2. Lines 27-28 should be changed to “Infection of WT mice with pathogenic but not acutely cleared CHIKV induced MyD88-dependent recruitment”.

3. In lines 189-190, the use of double negative is confusing and difficult to understand.

4. Line 237 should refer to Fig 6G.

Reviewer #3: Fig. 4L and M would be more informative with absolute levels demonstrated, rather than fold change which can be deceiving.

PLOS authors have the option to publish the peer review history of their article (what does this mean?). If published, this will include your full peer review and any attached files.

Reviewer #1: No

Reviewer #2: No

Reviewer #3: No

---

## [Decision Letter · Decision Letter 1]

22 Dec 2019

Dear Dr. Morrison, dear Dr. McCarthy, and fellow collaborators,

We are pleased to inform that your manuscript, "MyD88-dependent influx of monocytes and neutrophils impairs lymph node B cell responses to chikungunya virus infection via Irf5, Nos2 and Nox2", has been editorially accepted for publication at PLOS Pathogens. 

Before your manuscript can be formally accepted and sent to production, you will need to complete our formatting changes, which you will receive by email within a week. Please note that your manuscript will not be scheduled for publication until you have made the required changes.

IMPORTANT NOTES

(1) Please note, once your paper is accepted, an uncorrected proof of your manuscript will be published online ahead of the final version, unless you’ve already opted out via the online submission form. If, for any reason, you do not want an earlier version of your manuscript published online or are unsure if you have already indicated as such, please let the journal staff know immediately at plospathogens@plos.org.

(2) Copyediting and Proofreading: The corresponding author will receive a typeset proof for review, to ensure errors have not been introduced during production. Please review the PDF proof of your manuscript carefully, as this is the last chance to correct any errors. Please note that major changes, or those which affect the scientific understanding of the work, will likely cause delays to the publication date of your manuscript. 

(3) Appropriate Figure Files: Please remove all name and figure # text from your figure files. Please also take this time to check that your figures are of high resolution, which will improve the readbility of your figures and help expedite your manuscript's publication. Please note that figures must have been originally created at 300dpi or higher. Do not manually increase the resolution of your files. For instructions on how to properly obtain high quality images, please review our Figure Guidelines, with examples at: http://journals.plos.org/plospathogens/s/figures.

(4) Striking Image: Please upload a striking still image to accompany your article if one is available (you can include a new image or an existing one from within your manuscript). Should your paper be accepted, this image will be considered for our monthly issue image and may also appear on our website to feature your article. Please upload this as a separate file, selecting "striking image" as the file type upon upload. Please also include a separate "Other" file with a caption, including credits and any potential copyright information. Please do not include the caption in the main article file. If your image is from someone other than yourself, please ensure that the artist has read and agreed to the terms and conditions of the Creative Commons Attribution License at http://journals.plos.org/plospathogens/s/content-license. Please note that PLOS cannot publish copyrighted images.

(5) Press Release or Related Media: If your institution or institutions have a press office, please notify them about your upcoming paper at this point, to enable them to help maximize its impact. If they will be preparing press materials for this manuscript, please inform our press team in advance at plospathogens@plos.org as soon as possible. We ask that you contact us within one week to plan ahead of our fast Production schedule. If you need to know your paper's publication date for related media purposes, you must coordinate with our press team, and your manuscript will remain under a strict press embargo until the publication date and time. This means an early version of your manuscript will not be published ahead of your final version. 

(6)  PLOS requires an ORCID iD for all corresponding authors on papers submitted after December 6th, 2016. Please ensure that you have an ORCID iD and that it is validated in Editorial Manager.  To do this, go to ‘Update my Information’ (in the upper left-hand corner of the main menu), and click on the Fetch/Validate link next to the ORCID field.  This will take you to the ORCID site and allow you to create a new iD or authenticate a pre-existing iD in Editorial Manager

(7) Update your Profile Information: Now that your manuscript has been provisionally accepted, please log into Editorial Manager and update your profile, if needed. Go to https://www.editorialmanager.com/ppathogens, log in, and click on the "Update My Information" link at the top of the page. Please update your user information to ensure an efficient production and billing process. 

(8) LaTeX users only: Our staff will ask you to upload a TEX file in addition to the PDF before the paper can be sent to typesetting, so please carefully review our Latex Guidelines http://journals.plos.org/plospathogens/s/latex in the meantime.

(9) If you have associated protocols in protocols.io, please ensure that you make them public before publication to guarantee immediate access to the methodological details.

Best regards,

Marco Vignuzzi

Section Editor

PLOS Pathogens

Kasturi Haldar

Editor-in-Chief

PLOS Pathogens

orcid.org/0000-0001-5065-158X

Grant McFadden

Editor-in-Chief

PLOS Pathogens

orcid.org/0000-0002-2556-3526

Reviewer Comments (if any, and for reference):

Reviewer's Responses to Questions

**Part I - Summary**

Reviewer #1: I appreciate the author's efforts to address my comments and the other reviewer's concerns. I have no further concerns and am looking forward to the follow up studies.

Reviewer #2: The authors have nicely addressed all my concerns.

**Part II – Major Issues: Key Experiments Required for Acceptance**

Reviewer #1: (No Response)

Reviewer #2: (No Response)

**Part III – Minor Issues: Editorial and Data Presentation Modifications**

Reviewer #1: (No Response)

Reviewer #2: (No Response)

PLOS authors have the option to publish the peer review history of their article (what does this mean?). If published, this will include your full peer review and any attached files.

Reviewer #1: No

Reviewer #2: No

---

## [Editor Report · Acceptance letter]

22 Jan 2020

Dear Dr. Morrison,

We are delighted to inform you that your manuscript, "MyD88-dependent influx of monocytes and neutrophils impairs lymph node B cell responses to chikungunya virus infection via Irf5, Nos2 and Nox2," has been formally accepted for publication in PLOS Pathogens.

Best regards,

Kasturi Haldar

Editor-in-Chief

PLOS Pathogens

orcid.org/0000-0001-5065-158X

Michael Malim

Editor-in-Chief

PLOS Pathogens

orcid.org/0000-0002-7699-2064